# APEX: Empowering LLMs with Physics-Based Task Planning for Real-time Insight

## Abstract

Large Language Models (LLMs) demonstrate strong reasoning and task planning capabilities but remain fundamentally limited in physical interaction modeling. Existing approaches integrate perception via Vision-Language Models (VLMs) or adaptive decision-making through Reinforcement Learning (RL), but they fail to capture dynamic object interactions or require task-specific training, limiting their real-world applicability. We introduce APEX (Anticipatory Physics-Enhanced Execution), a framework that equips LLMs with physics-driven foresight for real-time task planning. APEX constructs structured graphs to identify and model the most relevant dynamic interactions in the environment, providing LLMs with explicit physical state updates. Simultaneously, APEX provides low-latency forward simulations of physically feasible actions, allowing LLMs to select optimal strategies based on predictive outcomes rather than static observations. We evaluate APEX on three benchmarks designed to assess perception, prediction, and decision-making: (1) Physics Reasoning Benchmark, testing causal inference and object motion prediction; (2) Tetris, evaluating whether physics-informed prediction enhances decision-making performance in long-horizon planning tasks; (3) Dynamic Obstacle Avoidance, assessing the immediate integration of perception and action feasibility analysis. APEX significantly outperforms standard LLMs and VLM-based models, demonstrating the necessity of explicit physics reasoning for bridging the gap between language-based intelligence and real-world task execution.

## 1 Introduction

A cat is about to pounce on an LLM-controlled agent. The agent detects the cat nearby and knows it should move, but does it understand that the cat will jump in 2 seconds? Once the LLM decides to evade, multiple escape routes exist, how does it choose a path that avoids both the cat and surrounding obstacles? These two challenges: **understanding dynamic interactions** and **predicting action consequences**, highlight fundamental limitations in existing LLM-based agents. Current methods attempt to address these issues using Vision-Language Models (VLMs) (Wang et al., 2024a; Ahn et al., 2022; Huang et al., 2024; 2023; Liang et al., 2023; Liu et al., 2024; Hu et al., 2023b) and Reinforcement Learning (RL) (Patel et al., 2025; Lee et al., 2024; Ma et al., 2024a; Sun et al., 2024). However, they remain fundamentally limited:

- **Static Perception Without Dynamic Awareness**: VLMs enable LLMs to recognize objects but fail to model interactions over time. They can detect a cat, but cannot anticipate its movement. In real-world decision-making, static snapshots are insufficient; understanding object motion is essential.
- **Lack of Action-Outcome Feedback and Physical Grounding**: Existing approaches often treat decision-making as a one-shot prediction task, offering no structured feedback loop between actions and their physical consequences. Instead of modeling the environment's response through grounded physical equations, they rely on latent dynamics (World Models) or reward-driven adaptation (RL). As a result, these systems lack interpretable quantitative feedback on the feasibility of action, e.g., whether an action would cause a collision, balance failure, or violate timing constraints.
- **Expensive and Slow Policy Adaptation**: RL-based approaches, such as VoxPoser (Huang et al., 2023) and Code-as-Policies (Liang et al., 2023), require extensive task-specific training. Every new scenario demands costly retraining, making real-time adaptation impractical.

Figure 1: Comparison of physical reasoning capabilities across three systems, LLM without spatial grounding, VLM and world modeling, and our proposed APEX on three scenarios involving object prediction, agent-object interaction, and action planning. While vanilla LLMs are not necessarily making random choices in the prediction task, our experimental results in Section 4 indicate that their performance is statistically indistinguishable from random selection in this context. APEX provides not only qualitative predictions but also quantitative estimations of outcomes (e.g., time to impact, risk of collision), demonstrating its structured understanding of physical causality.

To plan actions in the real world, agents must do more than perceive and react. They must simulate, quantify, and foresee. We introduce APEX (Anticipatory Physics-Enhanced Execution), a framework that enables LLMs to anticipate environmental changes and optimize actions through physics-based reasoning. APEX constructs structured graphs that extract the most relevant dynamic interactions in an environment(Nishida et al., 2018; Huang et al., 2025), enabling LLMs to reason about the motion and forces of objects. Additionally, APEX performs future state simulation(Smith et al., 2013), predicting how different actions will alter the environment over time, providing explicit physical constraints to guide decision-making. This strengthens the standard LLM's capabilities in physical reasoning, perception, and prediction, empowering LLM-driven agents to perform low-latency planning in physical environments, as illustrated in Fig. 1.

We evaluate APEX across three benchmark tasks; each is designed to address a critical limitation in existing approaches:

- **Physics Reasoning Benchmark (Addressing Static Perception)**: Testing LLMs' ability to infer object dynamics beyond simple object recognition.
- **Tetris (Evaluating Physics-Driven Foresight)**: Testing whether providing forward physical simulations as feedback improves the long-horizon decision quality of language models in structured planning environments.
- **Dynamic Obstacle Avoidance (Addressing Real-Time Adaptation)**: Assessing real-time integration of perception and prediction for adaptive decision-making, ensuring LLMs can dynamically adjust their behavior based on future state simulations.

We aim to close the gap between language-based reasoning and physically grounded execution. Our contributions are:

- **APEX**: a unified framework that equips LLMs with real-time perception with graph networks and physical foresight for dynamic task planning.
- A **three-part benchmark suite** spanning structured reasoning, long-horizon planning, and real-time control, each targeting a distinct dimension of physical intelligence.
- Empirical results showing that APEX outperforms LLMs and VLM-based agents in (1) numerical reasoning and physical calculation (Physics QA); (2) simulation-guided planning with physical intuition (Tetris); and (3) perception-integrated prediction for real-time decision-making (dynamic obstacle avoidance).

## 2    RELATED WORK

Despite significant progress in task planning for LLM-based or VLM-based agents (Wang et al., 2024b; Ma et al., 2024b; Kawaharazuka et al., 2024; Hu et al., 2023a), existing paradigms largely fail to integrate real-time physical modeling in embodied intelligence. Our work is situated at a unique intersection of language reasoning, graph-based physical abstraction, and online physics simulation.

### 2.1    VISION-LANGUAGE MODELS: PERCEPTION WITHOUT PHYSICAL CONSEQUENCE

Vision–language models (VLMs) such as CLIP (Radford et al., 2021), Flamingo (Alayrac et al., 2022), PaLM-E (Driess et al., 2023), and OpenVLM (Kim et al., 2024) learn powerful image–text embeddings that support zero-shot recognition and instruction following (Ma et al., 2024b). Many VLM-empowered agents, such as CLIPort (Shridhar et al., 2022), VIMA (Jiang et al., 2022), Vox-Poser (Huang et al., 2023), RT-2 (Brohan et al., 2023), and PhysVLM (Zhou et al., 2025) inherit this same static worldview. They augment visual grounding with spatial transport layers, multimodal prompting, or feasibility masks, but still cannot generalize to novel dynamics and remain blind to explicit physical laws. A handful of works have tried to close the loop by training Transformer-based action predictors directly on VLM features, for example, RT-1 (Brohan et al., 2022) learns end-to-end vision-to-control policies. DeepMind's generalist agent Gato (Reed et al., 2022) showed that a single Transformer can handle images, text, and control signals in a unified framework. Yet these approaches still encode physics only implicitly in learned weights, offering no transparent physical feedback and often failing under distributional shifts.

### 2.2    WORLD MODELS: FORESIGHT WITHOUT GUARANTEES

TWM (Robine et al., 2023) incorporates temporal attention into latent rollouts; SMART (Sun et al., 2023) adds self-supervised multi-task pretraining for control; R3M (Nair et al., 2022) leverages universal visual representations; and Genie (Bruce et al., 2024) integrates interactive environment generation. These models introduce video representation learning (Majumdar et al., 2023), multi-agent dynamics, and forward/inverse prediction, yet all remain black-box latent estimators without explicit guarantees of physical consistency. Despite their imaginative capabilities, latent world models exhibit fundamental limitations, including compounding roll-out errors that exacerbate over extended horizons, poor robustness to distributional shifts, and opaque latent dynamics that obscure failure modes and hinder interpretability. Furthermore, their temporal abstraction often distorts real physical intervals by embedding time into latent structures rather than modeling it explicitly.

### 2.3    REINFORCEMENT LEARNING: EXPENSIVE MASTERY, POOR GENERALIZATION

Reinforcement learning (RL) algorithms such as Proximal Policy Optimization (PPO) (Schulman et al., 2017) and Soft Actor–Critic (SAC) (Haarnoja et al., 2018), and Imitation Learning (IL) like Generative Adversarial Imitation Learning (GAIL) (Ho & Ermon, 2016) have achieved notable success in robotic control through extensive trial-and-error interaction. More recent LLM-guided RL hybrids aim to mitigate these issues by combining language reasoning with policy learning. SayCan (Ahn et al., 2022) uses a language model to rank actions proposed by a pretrained policy, and Inner Monologue (Huang et al., 2022) adds on-the-fly replanning via chain-of-thought prompts. Iker (Patel et al., 2025) augments low-level controllers with iterative keypoint rewards from a VLM. Models such as RT-1 Brohan et al. (2022) and BC-Z Jang et al. (2022) demonstrate the potential of large Transformer policies to generalize across multiple tasks after extensive pretraining on diverse environments. ProgPrompt (Singh et al., 2023), PromptCap (Hu et al., 2023c), and ECOt (Zawalski et al., 2024), chain LLM reasoning for task planning. Despite these advancements, RL and its LLM-centric extensions still face substantial challenges. They either require millions of environment steps to converge, struggle under physical distribution shifts, or rely on predefined controllers with limited adaptability to new physics interactions.

### 2.4    PHYSICAL SIMULATION IN LLM REASONING: BEYOND CONCEPTUAL HALLUCINATIONS

Prior works like Mind's Eye (Liu et al., 2022) and PiLoT (Zhang et al., 2023) propose injecting simulation-derived hints into LLM prompts to correct conceptual hallucinations, such as misunderstandings of qualitative physics (e.g., "heavier objects fall faster"). While effective for symbolic reasoning, these methods overlook a key dimension: **numerical precision**. Our experiments show that modern LLMs (e.g., GPT-4o(Achiam et al., 2023)) already grasp qualitative physical rules,

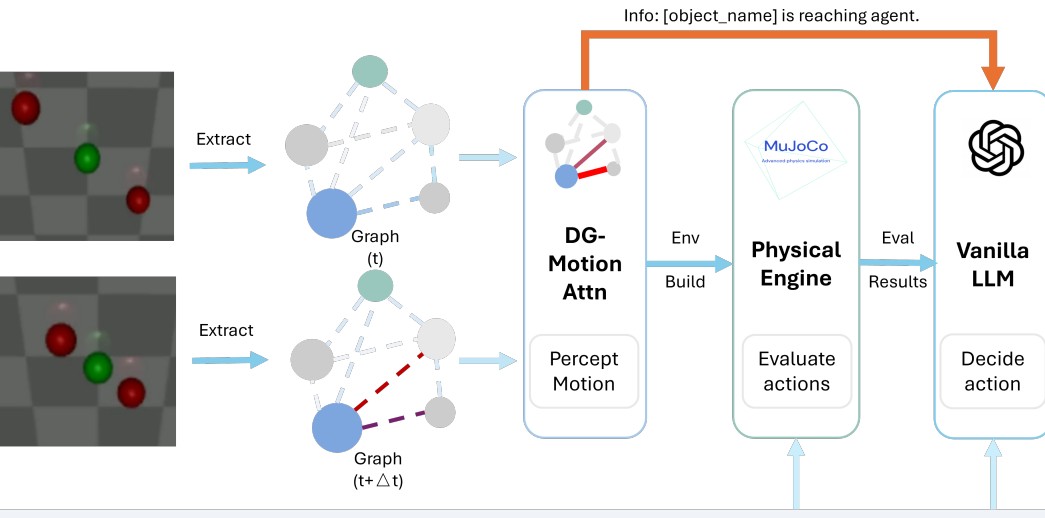

Figure 2: Overview of the APEX reasoning pipeline. Environment snapshots are abstracted into a motion-aware interaction graph via DG-Motion Attention. This graph structure triggers simulation in a physical engine (MuJoCo), which evaluates the outcome of candidate actions. A vanilla LLM then selects an action based on the simulated consequences. This loop, perception → graph trigger → simulation → LLM → action, enables grounded, temporally aware physical reasoning.

limiting the value of such corrections. However, they still fail at quantitative tasks, like predicting collision timing, unless grounded by external computation. In real-world environments where timing and magnitude are critical, this gap is consequential.

Table 1: Comparison of planning paradigms in dynamic physical environments.

| Method | Quant. Physics | Foresight | Resp. Time | Space–Time (Big-$O$) | Zero-shot |
|---|---|---|---|---|---|
| Vanilla LLMs | None | Implicit | Low | [1]$\mathcal{O}(p\,n)$ | Partial |
| VLMs | None | Implicit | Low | [2]$\mathcal{O}(p\,n)$ | Partial |
| World Models | Implicit | Latent rollout | Low–High | [3]$\mathcal{O}(h\,k\,p\,n)$–[3]$\mathcal{O}(h\,k\,p\,n^2)$ | Partial |
| RL / IL | None | Implicit | Low (infer) | train [4]$\mathcal{O}(s\,h\,p)$, infer [4]$\mathcal{O}(p)$ | No |
| **APEX (ours)** | **Explicit** | **Physics rollout** | **Low** | [5]$\mathcal{O}(h\,k\,n)$ | **Yes** |

 Resp. Time = per-decision inference latency (p95 bins: Low $\leq$ 2s; Medium 2–10s; High 10–60s; Very High $>$ 60s). $p$ = parameter count of models.
[1] $n$ = number of objects/state tokens per decision; no explicit lookahead $\Rightarrow$ near-linear cost.
[2] Cost dominated by perception (encode/decode once per decision); no multi-step rollout.
[3] $h$ = lookahead steps; $k$ = action space samples; worst-case $\mathcal{O}(h\,k\,n^2)$ with dense pairwise interactions; $\mathcal{O}(h\,k\,n)$ if sparsified.
[4] $s$ = training environment steps (high sample complexity); inference scales with $p$.
[5] Graph filtering reduces effective edges to $j \sim \mathcal{O}(p\,n)$; engine rollout $\mathcal{O}(h\,k\,n)$.
 **Zero-shot** means performing in *unseen* scenes/tasks/dynamics *without* fine-tuning; labels: *Yes* (robust), *Partial* (degrades but usable), *No* (requires adaptation).

## 3 METHODOLOGY

In this section, we introduce the APEX framework, structured explicitly in five detailed stages as shown in Fig. 2, systematically integrating physical reasoning into LLM decision-making. At its core, APEX leverages a graph-based representation explicitly chosen for its inherent ability to model relationships, not merely to highlight the most immediate or obvious actions but rather to capture complex, task-relevant interactions comprehensively.

## 3.1 GRAPH: RELATIONAL SCENE REPRESENTATION

Given consecutive snapshots at times $t$ and $t + \Delta t$, we construct relational graphs $G_t$ and $G_{t+\Delta t}$ over the same set of object nodes. Each node corresponds to a distinct entity in the environment, and edges encode potential interactions between pairs of objects. This relational graph structure explicitly represents the complex web of interactions, emphasizing task-relevant relationships rather than isolated physical states.

Such graph representations can be directly connected to upstream 3D reconstruction modules, serving as an intermediate abstraction layer between raw perceptual input and structured physical reasoning.

## 3.2 TRIGGER: DIFFERENCE-GRAPH MOTION ATTENTION

We form a *difference graph*:
$$\Delta G = G_{t+\Delta t} - G_t,$$
whose edges encode per-pair displacement, relative velocity, and newly emerging or evolving relationships. A Graphormer encoder computes attention scores:
$$\alpha_{ij} = \mathrm{Graphormer}(G_t, G_{t+\Delta t})_{ij},$$
identifying the most task-relevant edges based on relational dynamics. The selected edges define a focused subgraph, which is translated into a concise natural-language summary $S$, explicitly describing critical interactions and relationships (e.g., "sphere A is about to collide with B, influencing agent strategy").

## 3.3 SIMULATE: PHYSICS-GROUNDED ACTION ROLLOUTS

From the current relational state $s_t$, we enumerate a discrete set of candidate actions $\{a_i\}$ (e.g., left, right, down, jump). Note that while we define the action set $A$ as a collection of potential actions, the size of $A$ remains limited due to the finite degrees of freedom in current robotic systems, making enumeration feasible (Glover, 2004; Sutton et al., 1998). For each candidate action $a_i$, we invoke forward simulations:
$$s_{t+1}^{(i)} = \mathrm{PhysicsSim}(s_t, a_i),$$
and generate outcome descriptors $r_i$ (collision flags, target distances, object positions, durations). These outcome descriptors offer explicit, physics-grounded feedback tied directly to relational predictions and task implications.

## 3.4 LLM: GUIDED DECISION SYNTHESIS

We enrich the original LLM prompt $x$ with the relational summary $S$ and detailed simulation outcomes $\{r_1, \ldots, r_n\}$, resulting in a contextually comprehensive prompt:
$$x' = x \cup S \cup \{r_i\}.$$

The augmented context guides the LLM to synthesize the optimal relationally-informed decision sequence $\Pi$, representing a series of actions strategically selected to achieve the target objective based on predictive outcomes:
$$\Pi' = \arg\max_{\Pi} P_{\mathrm{LLM}}(\Pi \mid x').$$

## 3.5 ACT: EXECUTION OF THE OPTIMAL PLAN

The action plan $\Pi'$ is executed in the environment, realizing robust interactions grounded explicitly in relationally-informed physical foresight.

## 3.6 REPLACEMENT OF MODELS

APEX is modular. Each component can be replaced by a drop-in alternative as long as the *interfaces* are respected.

**Graph Trigger.** Any message-passing GNN or graph transformer that produces edge saliency scores over $(G_t, G_{t+\Delta t})$ is compatible, provided it yields a ranked set of task-relevant edges and a compact, textual summary $S$ for the current frame. Training and alternative encoders are detailed in Appendix 6.7.

**Physics simulator / world model.** PhysicsSim may be any engine capable of forward rollout (e.g., MuJoCo, Bullet, Brax) or a learned world model with bounded rollout error. The only requirement is to expose next-state predictions and outcome descriptors $r_i$ (collisions, distances, durations). Engine selection and learned-model variants are discussed in Appendix 6.8.

**Action search and complexity.** The default action set $A$ is small and enumerated. Time complexity and swap-in planners are summarized in Appendix 6.9.

---

**Algorithm 1** APEX: Anticipatory Physics-Enhanced Execution

---

**Require:** Environment snapshots at $t$ and $t + \Delta t$, LLM prompt $x$
**Ensure:** Final LLM-generated action plan $\Pi'$
 1: Construct relational graphs $G_t = (V, E)$ and $G_{t+\Delta t}$ from object states
 2: Compute attention scores via Graphormer:

$$\alpha_{ij} = \text{Graphormer}(G_t, G_{t+\Delta t})_{ij}$$

 3: Identify top-$k$ relationally salient edges forming focused subgraph $\tilde{G}$
 4: Generate summary $S$ from relational interactions within $\tilde{G}$
 5: Enumerate feasible actions $\{a_1, \ldots, a_n\}$ from current relational state
 6: **for** each action $a_i$ **do**
 7:     Simulate future state: $s_{t+1}^{(i)} = \text{PhysicsSim}(s_t, a_i)$
 8:     Generate outcome description $r_i = \text{Describe}(s_{t+1}^{(i)})$
 9: **end for**
10: Append summary $S$ and outcomes $\{r_1, \ldots, r_n\}$ to LLM prompt, forming enriched prompt $x'$
11: Decode optimal action plan from LLM:

$$\Pi' = \arg\max_{\Pi} P_{\text{LLM}}(\Pi \mid x')$$

12: **return** $\Pi'$

---

## 4 EXPERIMENTS

To evaluate APEX, we introduce a new **LLM Physical Reasoning Benchmark**, testing AI models' ability to predict and adapt to dynamic environments. The evaluation consists of three primary experiments as shown in Table 2. Results for other open-source LLMs are reported in Tables 8–13. Additional evaluations include a dedicated physical benchmark (Appendix 6.3) and a real-world application case study (Appendix 6.4).

Table 2: Summary of Experimental Setups and Physical Reasoning Capabilities

| Experiment | Capability Verified | Evaluation Objective |
|---|---|---|
| Physics Reasoning | Physical reasoning over multiple entities | Test LLM's ability to understand motion-related quantities across targets. |
| Tetris Planning | Foresight via simulated prediction | Assess whether physics-informed feedback improves planning quality. |
| Obstacle Avoidance | Perception-integrated prediction | Validate perception-action grounding under dynamic environments. |

### 4.1 EXPERIMENT 1: PHYSICAL REASONING ACCURACY IN STRUCTURED TASKS

To assess the physical reasoning capabilities of LLMs, we construct a suite of synthetic 3D tasks grounded in classical mechanics, including linear motion, circular motion, projectile motion, multi-object interactions, and collision prediction. Each task is framed as a structured reasoning problem: given object positions, velocities, and physical parameters, the LLM must infer whether collisions will occur or predict resulting velocities after interaction.

We compare vanilla GPT-4o against our APEX-enhanced GPT-4o in Table 3 and report three metrics:

- Accuracy: Whether the model provides a fully correct structured answer within the tolerance of 5%.
- Mean Squared Error (MSE): Quantitative deviation from ground-truth numerical values.
- Numerical Validity: Percentage of fields where the model returns valid numbers.

We conduct ablation experiments on different $dt$ in the physical simulation engine with the Euler forward method in Table 4. (Here, $dt$ refers to the step size in the physics engine's forward simulation, not the time interval in the Graph Trigger module.)

Table 3: Comparison of GPT-4o vs. APEX-enhanced GPT on Physical Reasoning Tasks. Across all five categories (linear, circular, projectile, collision, and multi-object motion), APEX achieves near-perfect accuracy, drastically lower MSE, and full numerical validity, while vanilla GPT-4o struggles on multi-object tasks.

| Task Type | Accuracy (%) ↑ | MSE ↓ | Numerical Validity (%) ↑ |
|---|---|---|---|
| *GPT-4o* | | | |
| 3D Linear Motion | 8.00 | 213.5931 | 28.00 |
| 3D Circular Motion | 24.00 | 4.0998 | 76.00 |
| 3D Projectile Motion | 88.00 | 303.6022 | 100.00 |
| 3D Collision | 44.00 | 12.4816 | 100.00 |
| 3D Multi-Object Motion | 0.00 | 1918.2065 | 81.33 |
| *APEX (ours)* | | | |
| 3D Linear Motion | **96.00** | **0.0076** | **100.00** |
| 3D Circular Motion | **100.00** | **0.0000** | **100.00** |
| 3D Projectile Motion | **100.00** | **0.0001** | **100.00** |
| 3D Collision | **88.00** | **2.4627** | **100.00** |
| 3D Multi-Object Motion | **97.33** | **0.0013** | **100.00** |

Table 4: Simulation accuracy and average duration per question type at different timesteps $dt$. Smaller timesteps ($dt = 0.001$) achieve the highest accuracy but incur longer runtimes, while larger timesteps ($dt = 0.010$) reduce computation at the cost of accuracy.

| Question Type | $dt = 0.001$ | | $dt = 0.005$ | | $dt = 0.010$ | |
|---|---|---|---|---|---|---|
| | Accuracy (%)↑ | Duration (s)↓ | Accuracy (%)↑ | Duration (s)↓ | Accuracy (%)↑ | Duration (s)↓ |
| 3D Linear Motion | 100.00 | 0.023 | 100.00 | 0.0058 | 96.00 | 0.0042 |
| 3D Circular Motion | 100.00 | 0.028 | 100.00 | 0.0068 | 96.00 | 0.0046 |
| 3D Projectile Motion | 92.00 | 0.013 | 92.00 | 0.0035 | 48.00 | 0.0027 |
| 3D Multi-Object Motion | 97.33 | 0.076 | 90.67 | 0.022 | 80.00 | 0.013 |
| 3D Collision | 98.00 | 0.0073 | 98.00 | 0.0090 | 98.00 | 0.0080 |

## 4.2 EXPERIMENT 2: REAL-TIME PHYSICAL PLANNING IN TETRIS

We design a second benchmark to test the agent's ability to perform dynamic, physics-informed planning in a classic block-stacking domain: Tetris. Unlike traditional planning tasks that focus on symbolic correctness or visual alignment, this environment emphasizes physical feasibility, spatial reasoning, and long-horizon optimization.

The agent interacts with a Tetris simulator in which it must select actions (left, right, rotate, drop) for falling blocks. Each decision must be made based on the current board state, the shape of the block, and the anticipated physical consequences of different placements. All models are run under the same sequence of five randomized seeds, and each episode is capped at 15 blocks with the estimated maximum number of clear lines as 3, ensuring fair and bounded comparison.

We compare different decision systems:

- **GPT-4o & GPT-4o-mini**: baseline LLMs with no physical modeling.

- **VLM**: GPT-4o with images as the VLM (Patel et al., 2025; Wang et al., 2025) that perceives the current board state via screenshot input.
- **APEX (ours)**: physical planning with physics-based rollout.

We evaluate each model on five physically grounded metrics:

- **Final Score**: total score after game termination (each cleared line counts 100).
- **Max Height**: the tallest column reached during gameplay.
- **Hole Count**: number of empty cells beneath landed blocks.
- **Bumpiness**: total height difference between adjacent columns.
- **Height Increase per Move**: average vertical growth rate per action.

These metrics reflect task performance and physical efficiency jointly. A low bumpiness and hole count indicate stable and compact stacking, while a lower height delta per move demonstrates the agent's foresight in minimizing vertical sprawl.

Table 5: Comparison of baselines vs. APEX on Tetris-style structured planning. Baseline models (GPT-4o, GPT-4o-mini, VLM) fail to clear lines and yield unstable, high stacks with many holes and bumps, whereas APEX achieves a large positive score with low stack height and smooth structure.

| Model | Final Score ↑ | Max Height ↓ | Holes ↓ | Bumps ↓ |
|---|---|---|---|---|
| GPT-4o | 0.0 | 14.6 | 33.4 | 25.6 |
| GPT-4o-mini | 0.0 | 18.2 | 26.0 | 36.4 |
| VLM | 0.0 | 12.6 | 30.2 | 22.6 |
| **APEX (ours)** | **140.0** | **5.0** | **2.8** | **6.8** |

### 4.3 EXPERIMENT 3: DYNAMIC OBSTACLE AVOIDANCE

This experiment assesses the agent's adaptive decision-making capabilities within dynamic physical environments characterized by moving obstacles. The setup utilizes a simulated MuJoCo environment where an LLM-driven agent navigates through varying obstacle densities and speeds across different difficulty levels.

The evaluation metrics are as follows:

- **CFR (Collision-Free Rate)**: the rate of time in which the agent successfully avoids all obstacles.
- **IAR (Invalid Action Rate)**: the proportion of actions that lead to collisions or unsafe states.
- **AST (Average Survival Time)**: the average duration the agent remains operational without colliding, reflecting overall navigation efficacy.

Table 6: Performance on real-time obstacle avoidance across task complexities. Baselines (GPT-4o, GPT-4o-mini, VLM) fail to generalize beyond trivial cases, yielding near-zero success rates. By contrast, APEX consistently achieves high completion rates with zero invalid actions across all settings, maintaining robust performance even as task complexity increases.

| Model | Simple | | | Medium | | | Hard | | |
|---|---|---|---|---|---|---|---|---|---|
| | CFR ↑ | IAR (%)↓ | AST (s)↑ | CFR ↑ | IAR (%)↓ | AST (s)↑ | CFR ↑ | IAR (%)↓ | AST (s)↑ |
| GPT-4o-mini | 0/5 | 0 | 2.55 | 0/5 | 0 | 1.86 | 0/5 | 0 | 2.24 |
| GPT-4o | 1/5 | 0 | 5.85 | 0/5 | 0 | 3.15 | 0/5 | 0 | 1.66 |
| VLM | 0/5 | 7 | 5.18 | 0/5 | 4 | 3.14 | 0/5 | 7 | 2.48 |
| **APEX (GPT-4o-mini)** | **5/5** | **0** | **10.00** | **3/5** | **0** | **8.64** | **1/5** | **0** | **6.86** |
| **APEX (GPT-4o)** | **5/5** | **0** | **10.00** | **5/5** | **0** | **10.00** | **3/5** | **0** | **8.07** |

We conduct ablation experiments on different graph models and different values of $k$, as reported in Table 7. The choice of $k$ controls how many relational edges are passed forward after motion-based saliency filtering: small $k$ may discard critical interactions, while large $k$ increases noise and computational overhead. Similarly, the graph encoder defines how relational dynamics are aggregated; we compare GAT, GCN, and Graphormer to evaluate whether higher-order attention mechanisms improve action planning performance. This ablation isolates the contribution of edge selection ($k$) and relational modeling capacity (graph backbone) to overall system performance.

Table 7: Ablation study on hard obstacle avoidance: Top-$k$ selection vs. graph model choice. Performance is highly sensitive to both hyperparameters: $k = 2$ with GPT-4o provides the best trade-off in success rate and planning stability, while Graphormer shows moderate gains over GAT/GCN. Mini variants fail across all settings, underscoring the need for both sufficient LLM capacity and structured graph filtering.

| Ablation | $k$/Model | LLM | CFR↑ | AST (s)↑ | IAR(%)↓ | Latency (s)↓ |
|---|---|---|---|---|---|---|
| | $k = 1$ | gpt-4o-mini | 0/5 | 7.17 | 0 | 0.74 |
| | $k = 2$ | gpt-4o-mini | 0/5 | 6.38 | 0 | 0.73 |
| Top-$k$ | $k = 4$ | gpt-4o-mini | 0/5 | 7.53 | 0 | 0.74 |
| | $k = 1$ | gpt-4o | 0/5 | 4.97 | 0 | 0.94 |
| | $k = 2$ | gpt-4o | 2/5 | 6.58 | 0 | 1.25 |
| | $k = 4$ | gpt-4o | 2/5 | 5.58 | 0 | 1.29 |
| | GAT | gpt-4o-mini | 0/5 | 2.85 | 0 | 0.89 |
| | GCN | gpt-4o-mini | 0/5 | 7.08 | 0 | 0.85 |
| Graph Model | Graphormer | gpt-4o-mini | 0/5 | 7.59 | 0 | 0.76 |
| | GAT | gpt-4o | 0/5 | 4.99 | 0 | 0.62 |
| | GCN | gpt-4o | 0/5 | 5.26 | 0 | 0.70 |
| | Graphormer | gpt-4o | 1/5 | 5.44 | 0 | 1.24 |

## 4.4 EVALUATION SUMMARY

APEX substantially augments LLM capabilities in physical reasoning across structured tasks, dynamic adaptation, and real-time obstacle avoidance. Our findings indicate that APEX consistently outperforms standard LLMs, achieving over 90% accuracy in multi-object dynamics (Experiment 1), efficient long-horizon planning (Experiment 2), and proactive collision avoidance (Experiment 3).

In Experiment 1, APEX demonstrates superior accuracy in predicting circular motion and collision dynamics, with baseline GPT-4o achieving less than 20% in Table 3.

In Experiment 2 (Tetris), APEX leverages predictive foresight to minimize structural irregularities, optimizing placements and significantly improving task performance in Table 5.

Experiment 3 further underscores APEX's advantage in real-time obstacle avoidance, effectively mitigating collision risks through predictive modeling, a critical gap in baseline GPT and VLM systems in Table 6.

## 5 CONCLUSION

In this paper, we introduced APEX, a novel framework that enhances LLMs with predictive physical reasoning capabilities by integrating graph-based physical modeling, and physics simulation. Unlike prior methods that rely on static observations or constraint filtering, APEX enables LLMs to anticipate future physical interactions and adapt task plans accordingly. Experimental results demonstrate that APEX significantly improves performance on physical reasoning benchmarks, outperforming standard LLMs, VLM-based task planning, and grounded decoding techniques.

Furthermore, APEX's structured approach to physical modeling opens new opportunities for future research in AI-driven task planning, robotics, and autonomous decision-making. This study provides a new perspective on enhancing LLMs' physical reasoning capabilities by replacing RL-based trial-and-error learning with predictive physical modeling. This direction presents new possibilities for future robotic task planning and can be combined with existing VLM+RL-based methods to further improve LLMs' ability to handle physical interaction tasks.

## 6 FUTURE WORK

As a next step, we aim to extend APEX into APEX++, where the language model serves not only as a planner, but as a core component in a recurrent, interpretable perception-prediction-action loop. This would allow for the emergence of grounded intelligence capable of proactive behavior, structured foresight, and physical adaptability, unlocking new possibilities across robotics, self-driving, and embodied AI.

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

# Appendix

## 6.1 DESIGN PRINCIPLE: PERCEPTION–GRAPH–LANGUAGE–PHYSICS–ACTION (PGLPA) VS. VISION–LANGUAGE–ACTION (VLA)

A key design principle underlying our framework diverges from the conventional *Vision–Language–Action* (VLA) paradigm, **to connect real-to-sim-to-real with LLM reasoning, while keeping the blackbox models isolated from numerical/physical information.** We refer to our modular approach as *Perception–Graph–Language–Physics–Action* (PGLPA). Figures 3 and 4 contrast the conventional VLA pipeline with our proposed PGLPA paradigm, highlighting the structural differences that motivate our approach. We further compare the two paradigms in terms of accuracy and hallucination, training and data requirements, and interpretability as follows.

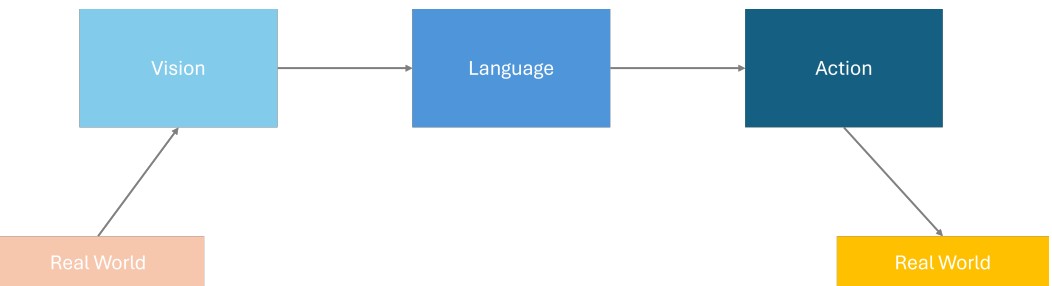

Figure 3: Illustration of the conventional Vision–Language–Action (VLA) paradigm. Visual perception encodes the real world into language features, which are then mapped directly to action commands. The execution loop closes by applying actions back to the real world. Although conceptually simple, VLA tightly couples perception, reasoning, and control within a single embedding space, limiting interpretability and robustness.

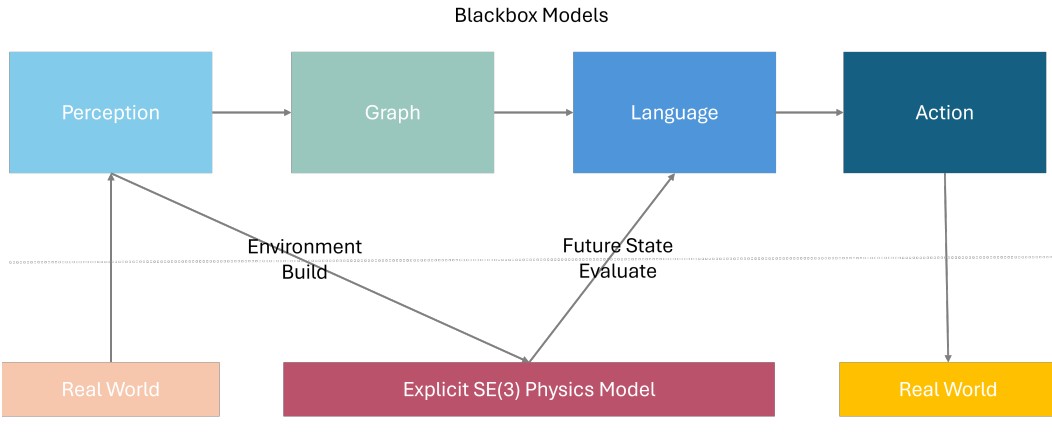

Figure 4: Illustration of our Perception–Graph–Language–Physics–Action (PGLPA) paradigm. Perception constructs a relational graph from the real world; this graph informs both symbolic reasoning and an explicit SE(3)-consistent physics simulator. The simulator evaluates candidate actions via rollouts, producing structured feedback that is integrated with LLM-based reasoning before execution. This "real-to-sim-to-real" loop decouples numerical physical computation from probabilistic inference, improving stability, interpretability, and zero-shot transfer.

**Accuracy and Hallucination.** Planning in dynamic scenes is inherently spatio-temporal: agents must reason over objects' states and transitions in 4D under the constraints of Newtonian mechanics. With *full* observation, physical laws enable forward prediction of future states; the residual difficulty comes from action selection, which is combinatorially hard (often treated as a weakly NP-type search problem; see e.g., classical results on planning/search). Under *partial* observation, priors (e.g.,

plausible mass ranges) and online updates (e.g., quick weighing) are required to reduce uncertainty. Modern deep models, including transformers, approximate such unknowns probabilistically; however, their physical modeling is *implicit*, which leads to two issues: (i) numerical instability for arithmetic operations[1], and (ii) lack of strict SE(3) consistency (viewpoint changes can disrupt spatial constancy).

PGLPA addresses both by performing all physics in an *explicit*, SE(3)-consistent environment (sim/engine) and using perception/graph/LLM only for probabilistic inference and decision. Explicit physics also constrains VLM/LLM hallucinations, and solving partial observability in a structured physical model is substantially simpler than tackling it end-to-end in a monolithic VLA.

**Training and Data Requirements.** VLA typically demands joint *vision×language×action* datasets and end-to-end training. In contrast, PGLPA trains mature modules independently and composes them via stable interfaces. Adding a new modality (e.g., LiDAR) requires retraining only the perception module rather than the full stack. For action selection, we can leverage simulator-backed search (e.g., RL/MCTS/CEM) directly in the physics environment; this is generally more data-efficient than learning a vision→action mapping end-to-end, and aligns with simulator-to-real fine-tuning practices.

**Interpretability.** All PGLPA modules expose explicit outputs with clear supervision:
- The perception model performs object categorization and 2D/3D localization.
- The graph module acts as a filter, surfacing salient interactions (see Appendix roadmap for a definition of "salient").
- The LLM is pre-trained for commonsense and reasoning (task decomposition, option selection, safety-aware judgments).
- The physics module conducts forward and counterfactual rollouts; their combination enables capabilities beyond standard RL (e.g., "if I do not block at $(0, 0, 1)$, the car will hit the child, which is immoral and unaffordable.").
- The action module follows from the training discussion above.

---

[1]Transformers and related architectures are not reliable for exact arithmetic/iteration operations, especially exponential or iterative routines (Garg et al., 2022). It aligns with our results in Table 3.

## 6.2 SUPPLEMENTARY EXPERIMENTS ON MORE LLMS

We further conduct the three experiments on five recent LLMs, with detailed results presented in Tables 8 to 13.

Table 8: Generalization across LLM backbones - Physical QA. This table compares the accuracy and response latency of five recent LLMs across diverse physical reasoning tasks. While these models demonstrate varying capabilities, none surpass the accuracy achieved by our GPT-4o + APEX framework reported in Table 3.

| LLM | Linear | | Circular | | Projectile | | Multi Obj | | Collision | |
|---|---|---|---|---|---|---|---|---|---|---|
| | Acc (%)↑ | Time (s)↓ | Acc (%)↑ | Time (s)↓ | Acc (%)↑ | Time (s)↓ | Acc (%)↑ | Time (s)↓ | Acc (%)↑ | Time (s)↓ |
| GPT-4.1 | 52.00 | 3.767 | 44.00 | 4.120 | 92.00 | 3.093 | 12.00 | 4.723 | 28.00 | 8.170 |
| DeepSeek-R1* | 100.00 | 193.934 | 80.00 | 356.351 | 100.00 | 349.337 | 86.65 | 310.937 | 40.00 | 363.831 |
| Claude Sonnet 4 | 100.00 | 6.845 | 16.00 | 4.387 | 100.00 | 6.808 | 6.67 | 8.686 | 38.00 | 10.019 |
| Gemini 2.5 Flash | 80.00 | 10.593 | 40.00 | 7.434 | 92.00 | 11.766 | 32.00 | 19.168 | 70.00 | 58.761 |
| LLaMA 4 Scout | 0.00 | 6.141 | 4.00 | 5.583 | 72.00 | 6.687 | 1.33 | 5.770 | 10.00 | 5.824 |

For DeepSeek-R1, only 20% of the dataset was evaluated due to its significantly longer reasoning time, which made full-scale benchmarking impractical within our resource constraints.

Table 9: Performances of LLMs for the Tetris Experiment. Gemini achieved the best structural control with the lowest stack height, though its latency was very high. Claude and Gemini occasionally cleared lines and maintained moderate structure. GPT-4.1 was fast but structurally weak, while LLaMA failed all cases with rigid stacking behavior. Overall, Gemini appears to perform the best, achieving the lowest average max stack height (9.2±2.48). For reference, the APEX (GPT-4o) baseline maintains an average max height of 5±2.97.

| Model | Final Score↑ | Max Stack Height↓ | Holes↓ | Bumps↓ | Resp. Time (s)↓ |
|---|---|---|---|---|---|
| GPT-4.1 | $0.0 \pm 0.0$ | $15.0 \pm 2.61$ | $38.4 \pm 17.67$ | $27.4 \pm 7.36$ | $0.778 \pm 0.116$ |
| Claude Sonnet 4 (20250514) | $20.0 \pm 40.0$ | $14.4 \pm 0.80$ | $36.4 \pm 6.86$ | $17.8 \pm 3.92$ | $1.557 \pm 0.049$ |
| Gemini 2.5 Flash | $20.0 \pm 40.0$ | $9.2 \pm 2.48$ | $14.2 \pm 5.84$ | $13.6 \pm 5.68$ | $85.391 \pm 7.625$ |
| LLaMA 4 Scout | $0.0 \pm 0.0$ | $17.0 \pm 0.00$ | $32.2 \pm 5.46$ | $30.2 \pm 5.19$ | $0.852 \pm 0.054$ |

For clarity, we removed the "Height/move" metric, which was effectively redundant with max stack height as it was not normalized by the number of moves.

Table 10: Performances of LLMs for the Tetris Experiment with Vision. All models failed to clear lines with image input. Gemini maintained the lowest stack height but had high latency, Claude showed balanced structural metrics, GPT-4.1 was fast but unstable, and LLaMA consistently terminated at max height due to rigid behavior.

| Model | Final Score↑ | Max Stack Height↓ | Holes↓ | Bumps↓ | Resp. Time (s)↓ |
|---|---|---|---|---|---|
| GPT-4.1 | $0.0 \pm 0.0$ | $12.6 \pm 2.73$ | $24.4 \pm 9.05$ | $26.0 \pm 10.35$ | $1.162 \pm 0.096$ |
| Gemini 2.5 Flash | $0.0 \pm 0.0$ | $10.8 \pm 1.47$ | $23.2 \pm 7.19$ | $15.8 \pm 4.02$ | $47.490 \pm 8.011$ |
| LLaMA 4 Scout | $0.0 \pm 0.0$ | $17.6 \pm 1.36$ | $34.8 \pm 11.89$ | $35.2 \pm 2.71$ | $0.892 \pm 0.105$ |
| Claude Sonnet 4 (20250514) | $0.0 \pm 0.0$ | $12.2 \pm 2.23$ | $22.4 \pm 7.45$ | $21.6 \pm 8.96$ | $1.736 \pm 0.114$ |

Table 11: Performances of LLMs and LLM+Vision models on the Simple dynamic obstacle avoidance task. Claude and GPT-4.1 reached 80% task success, while DeepSeek and Gemini showed extremely long latencies.

| Type | Model | CFR↑ | AST (s)↑ | IAR (%)↓ | Avg Latency (s) ↓ |
|------|-------|------|----------|----------|-------------------|
| LLM+Vision | Claude Sonnet 4 (20250514) | 4/5 | 9.98 | 2.00 | 2.00 |
| LLM+Vision | Gemini 2.5 Flash | 1/5 | 6.73 | 4.00 | 27.45 |
| LLM+Vision | LLaMA 4 Scout | 1/5 | 5.24 | 0.00 | 1.27 |
| LLM+Vision | GPT-4.1 | 0/5 | 6.84 | 0.00 | 1.19 |
| LLM | DeepSeek-R1 (0528) | 0/5 | 2.55 | 7.00 | 88.08 |
| LLM | Claude Sonnet 4 (20250514) | 0/5 | 5.79 | 0.00 | 1.54 |
| LLM | Gemini 2.5 Flash | 0/5 | 6.64 | 7.00 | 2.36 |
| LLM | LLaMA 4 Scout | 0/5 | 6.83 | 0.00 | 0.88 |
| LLM | GPT-4.1 | 4/5 | 8.89 | 0.00 | 0.92 |

Table 12: Performances on the Medium difficulty setting. Most models failed to generalize. DeepSeek and Gemini still exhibited long planning times, while Claude and GPT-4.1 remained efficient.

| Type | Model | CFR↑ | AST (s)↑ | IAR (%)↓ | Avg Latency (s) ↓ |
|------|-------|------|----------|----------|-------------------|
| LLM+Vision | Claude Sonnet 4 (20250514) | 0/5 | 2.58 | 0.00 | 1.92 |
| LLM+Vision | Gemini 2.5 Flash | 0/5 | 4.47 | 4.00 | 28.07 |
| LLM+Vision | LLaMA 4 Scout | 0/5 | 5.35 | 0.00 | 1.24 |
| LLM+Vision | GPT-4.1 | 0/5 | 1.86 | 0.00 | 1.25 |
| LLM | DeepSeek-R1 (0528) | 0/5 | 2.43 | 13.00 | 179.01 |
| LLM | Claude Sonnet 4 (20250514) | 0/5 | 3.96 | 2.00 | 2.68 |
| LLM | Gemini 2.5 Flash | 0/5 | 2.53 | 11.00 | 33.03 |
| LLM | LLaMA 4 Scout | 0/5 | 4.28 | 0.00 | 0.82 |
| LLM | GPT-4.1 | 0/5 | 3.16 | 0.00 | 0.71 |

Table 13: Performances on the Hard setting. No model succeeded, but latency differences remained stark. Gemini and DeepSeek remain impractical for time-sensitive planning.

| Type | Model | CFR ↑ | AST (s)↑ | IAR (%)↓ | Avg Latency (s) ↓ |
|------|-------|-------|----------|----------|-------------------|
| LLM+Vision | Claude Sonnet 4 (20250514) | 0/5 | 4.63 | 2.00 | 2.14 |
| LLM+Vision | Gemini 2.5 Flash | 0/5 | 3.57 | 0.00 | 35.65 |
| LLM+Vision | LLaMA 4 Scout | 0/5 | 1.66 | 0.00 | 1.26 |
| LLM+Vision | GPT-4.1 | 0/5 | 3.04 | 0.00 | 1.23 |
| LLM | DeepSeek-R1 (0528) | 0/5 | 3.18 | 9.00 | 228.02 |
| LLM | Claude Sonnet 4 (20250514) | 0/5 | 3.55 | 0.00 | 1.65 |
| LLM | Gemini 2.5 Flash | 0/5 | 3.77 | 4.00 | 34.34 |
| LLM | LLaMA 4 Scout | 0/5 | 2.33 | 7.00 | 0.67 |
| LLM | GPT-4.1 | 0/5 | 4.20 | 0.00 | 1.42 |

## 6.3 SUPPLEMENTARY EXPERIMENTS ON THE PHYRE BENCHMARK

We additionally evaluate APEX on the Phyre benchmark (Bakhtin et al., 2019), a widely used suite of physical reasoning puzzles that require agents to anticipate object dynamics and plan interventions in diverse 2D environments. Each task is defined by a goal condition (e.g., make the green ball touch the blue box) and requires reasoning about gravity, collisions, and multi-object interactions. Unlike synthetic kinematics tests, Phyre emphasizes generalization: models must solve both seen and unseen templates, making it a strong proxy for zero-shot physical reasoning. This benchmark allows us to assess whether APEX's graph–simulation loop provides advantages in standardized tasks beyond our custom environments.

To simulate potential sim-to-sim or sim-to-real discrepancies in real-world settings, we implemented a disturbed simulator in the $256 \times 256$ environment: each object was perturbed with up to 2 pixels in position and $1°$ in rotation. We did not repeat standard sim-to-sim comparisons for two reasons: (1) time constraints, and (2) most RL agents, except those that explicitly address vision or sim-to-sim transfer, are trained in the original simulation environment and are not typically designed to generalize across simulators.

Table 14: GPT-4.1 nearly completely failed to solve the task. DeepSeek-R1 took a significantly long time ($\sim$150s per case) but still solved only a small number of problems. In contrast, our APEX-enhanced GPT-4.1, even under disturbed conditions, consistently produced valid rollouts and outperformed analytical methods by a wide margin.

| Model | Task Type | Total Tasks | Solved↑ | Solved (%)↑ | Avg Resp. Time (s)↓ | Avg Sim Time (s)↓ | AUCCESS↑ | Attempts / Task↓ |
|---|---|---|---|---|---|---|---|---|
| GPT-4.1 | ball_cross_template | 500 | 2 | 0.40% | 5.188 | 0.000 | 0.004 | 2.918 |
| GPT-4.1 | ball_within_template | 500 | 6 | 1.20% | 4.945 | 0.000 | 0.0114 | 2.958 |
| DeepSeek-R1 | ball_cross_template | 20 | 0 | 0.00% | 170.915 | 0.000 | 0.000 | 2.800 |
| DeepSeek-R1 | ball_within_template | 20 | 3 | 15.00% | 133.611 | 0.000 | 0.119 | 2.800 |
| APEX (GPT-4.1) | ball_cross_template | 500 | 261 | **52.20%** | 5.735 | 14.654 | 0.487 | 3.978 |
| APEX (GPT-4.1) | ball_within_template | 500 | 289 | **57.80%** | 5.689 | 12.181 | 0.542 | 3.826 |

All simulations were run sequentially on CPU without GPU or distributed computing. Parallelization would significantly reduce total runtime.
The action space was explored using 10,000 actions sampled uniformly at random.
Due to DeepSeek-R1's high inference cost, only 20 out of 500 test cases were evaluated.

## 6.4 SUPPLEMENTARY EXPERIMENTS ON REAL WORLD APPLICATION

We further validate APEX in a real-world robotic setting, using a reactive collision avoidance task.

**Experimental Setup.** The platform is a HiWonder Mini Arm (5 DOF) controlled by a Raspberry Pi 4B (Arm32, Python 3.7). An onboard RGB-D camera is mounted on the end-effector for visual input. Perception is implemented with classical CV techniques including color-based segmentation, bounding box tracking, and depth estimation, followed by a 5-frame sliding-window filter for position smoothing. The agent receives prompts in the format specified in the following. Baseline comparisons use GPT-4o directly, queried once per second without trigger or simulation.

**Prompt of Explicit intervention setting(APEX and GPT-4o)**

```
You are controlling a robot arm in a 2D tabletop environment.
Two balls are moving on the table: a red ball and a green ball.
The red ball is stationary, and the green ball is moving toward it.

Your task is to **prevent a collision** between them
by moving the robot arm to intercept the green ball.

Please choose a 2D target position (x, y),
where the robot arm should go to block the green ball's path.
The robot arm will then move to the position (x, y, 0.5) in 3D space,
hovering slightly above the table.
```

```
Make sure the chosen position is effective in preventing the collision,
but also avoid placing the robot arm too close to the red ball.

Current_state:{state}
Physical Engine Result: {rolling_results} (Prompt Injection)
Return your result as a JSON dictionary: {{"x": ..., "y": ...}}
or {{"x":-99, "y":-99}} if you think no need of action
Return Only The JSON without Markdown
```

**Prompt of Implicit intervention setting(APEX and GPT-4o)**

```
You are controlling a robot ball on 2D board.
It can stop any object near in any movement
You can move the ball to a location (x,y) in 1 sec
Current_state:{state}
The green car is reaching the child in red T-shirt in 5 sec.

Physical Engine Result: {rolling_results}

Return your result as a JSON dictionary: {{"x": ..., "y": ...}}
or {{"x":-99, "y":-99}} if you think no need of action
Return Only The JSON without Markdown
```

**Task.**  Explicit intervention setting: A human moves a green block toward a static red block. The agent must detect the potential collision and move the manipulator to prevent contact. Implicit intervention setting: In the same setting, but We do not explicitly tell the LLM that it needs to intervene in a collision. We only inform it that it controls a ball that can stop any object, and that a green car is approaching a kid in a red T-shirt from the graph model.

**Metrics.**  We measure response rate, collision rate, and planning latency.

Table 15: In the no-moving condition, we provide the LLM with the ball's position and velocity. When prompted to intervene, GPT-4o tends to react.

| Model | FIR↓ | Resp. Time (s)↓ |
|---|---|---|
| GPT-4o | 8/10 | 4.534 |
| APEX (GPT-4o) | 0/10 | – |

Table 16: In the collision condition, we evaluate the intervention behavior of GPT-4o and APEX-augmented GPT-4o in the same linear collision scenario. APEX significantly improves both the validity and success rate of interventions, while also reducing response time and simulation delay.

| Model | Resp. Rate↑ | Valid↑ | Success ↑ | Resp. Time (s)↓ | Sim Time (s)↓ |
|---|---|---|---|---|---|
| GPT-4o | 10/10 | 5/10 | 3/10 | 5.342 | – |
| APEX (GPT-4o) | 10/10 | 8/10 | 8/10 | 1.6562 | 0.1855 |

**Limitations**  Our deployment platform was a Raspberry Pi 4B (ARM32 architecture) with a system-level Python version restricted to 3.7. Under these constraints, PyTorch installation was infeasible. We therefore employed a lightweight linear classifier to estimate whether a selected object would collide with an obstacle within a 5-second horizon. This linear predictor can also serve as a pseudo-label generator for training a graph-based collision forecasting model.

In this hardware setting, Mujoco deployment was also not feasible. Given the simplicity of the task, we implemented a custom forward Euler integrator as a proxy simulator. For each object, trajectories

Table 17: Five consecutive trials for the implicit intervention setting. In this condition, we do not explicitly tell the LLM that it needs to intervene in a collision. The LLM only knows it controls a ball that can stop any object, and that a green car is approaching a kid in a red T-shirt from the graph model. Among the two failure cases: one was due to a simulation error where no feasible stopping point was found; the other was because the LLM did not respond and chose not to intervene.

| Model | Success↑ |
|---|---|
| APEX (GPT-4o) | 3/5 |

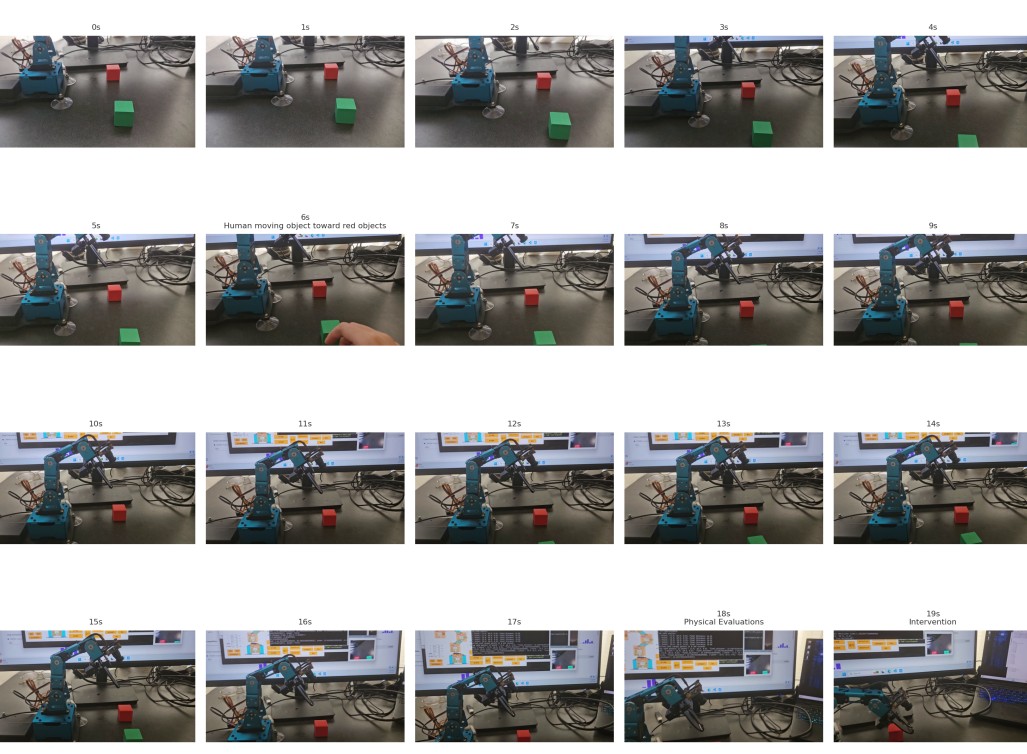

Figure 5: Frame montage from the real-world deployment video, with one frame sampled per second. The sequence illustrates three key phases of the experiment: (**6s**) human moving a green object toward a static red object, (**18s**) physical evaluations by the APEX simulation loop, and (**19s**) intervention performed by the robotic arm to prevent collision. This visualization highlights how APEX integrates perception, simulation, and LLM reasoning into grounded physical action.

```
Top safe points:
   Point: (1.0, 18.5, 0.0), Final Distance: 13.25
   Point: (1.5, 18.5, 0.0), Final Distance: 13.25
   Point: (2.0, 18.5, 0.0), Final Distance: 13.25
   Point: (2.5, 18.5, 0.0), Final Distance: 13.25
   Point: (0.5, 18.5, 0.0), Final Distance: 13.00

Rolled 1412 target points in 0.186 seconds.

Top safe points: [
   {'point': (1.0, 18.5, 0.0), 'final_distance': 13.246743148225988},
   {'point': (1.5, 18.5, 0.0), 'final_distance': 13.246743148225988},
   {'point': (2.0, 18.5, 0.0), 'final_distance': 13.246743148225988},
   {'point': (2.5, 18.5, 0.0), 'final_distance': 13.246743148225988},
   {'point': (0.5, 18.5, 0.0), 'final_distance': 13.003796642816358}
]
```

Figure 6: Filtered top-5 safe nodes from physical analysis from 1412 points in 0.186 seconds.

were computed using the first-order update:

$$\text{pos}[t+1] = \text{pos}[t] + v[t] \cdot \Delta t, \quad v[t+1] = v[t] + a[t] \cdot \Delta t.$$

Since Mujoco also defaults to Euler integration unless explicitly reconfigured with higher-order solvers, our approximation remains consistent with the default dynamics fidelity. On the Raspberry Pi 4B, simulating 1412 points over a 5-second window with $\Delta t = 0.01$s takes approximately 0.2s, which is negligible compared to LLM inference latency (1.5–5.0s).

## 6.5 PROMPT FORMATS AND MODEL INPUTS

To ensure consistency and replicability across models and tasks, we provide the exact prompt templates used in each experimental setting. All inputs are designed to maintain clarity while preserving the reasoning and response structure expected by LLMs.

**Physics QA Prompt Format (APEX and GPT-4o):**

```
You are a physics expert.(System Prompt)

Solve the following problem and return the answer in JSON format.

Problem: {q["question"]}

The external physical engine predictions: {ref} (Prompt Injection)

Expected JSON response:
{{
    "reasoning": "Explanation of how you arrived at the answer"
    "answer": "Your final numerical answer(without unit and equation)"
    as {str(q['answer_json'])},
}}

Respond the JSON string only without any markdown symbol.
```

**Tetris Planning Prompt Format (APEX and GPT-4o):**

```
You are a Tetris AI agent.(System prompt)

You are playing Tetris. Your goal is to maximize the score by:
- Clearing as many lines as possible.
- Keeping the board as flat as possible.
- Avoiding unnecessary stacking.

Here is the current board state(0-blank,,1-current piece, 2-landed piece):
{state}

Here are physical engine analysis:{APEX_results} (Prompt Injection)

Available moves:
- "left": Move the piece left by one column.
- "right": Move the piece right by one column.
- "rotate": Rotate the piece 90 degrees clockwise.
- "down": Instantly drop the piece to the lowest possible position.(max times = 1)

Decide the best move sequence in JSON format as a list of actions.
Each action should include the move and how many times to perform it.

Example:
[
  {{"move": "left", "times": 2}},
  {{"move": "rotate", "times": 1}},
  {{"move": "down", "times": 1}}
]

Allowed moves are: "left", "right", "rotate", and "down".
Only return the JSON array without any explanation or markdown. No Markdown
```

**Obstacle Avoidance Prompt (APEX and GPT-4o):**

```
You are an AI robot that avoids dynamic obstacles.(System Prompt)
You are controlling a robot in a 3D physical environment with moving obstacles.
Your goal is to avoid collisions with cats while progressing toward the target
location.

Current state
(The map has square walls located at x = ±5 meters and y = ±5 meters):
{state}

Obstacles:
{summary}

Available Moves:
{available_move}

Physical Engine Analysis:
{apex_results} (Prompt Injection)

Output the decision in this format:
{{
"move": "stay",
"duration": 1.0,
}}

Only return the JSON object with no explanation or markdown.

Here is the screenshot
(Red balls cat, green ball-your controlled agent): {image} (VLM only)
```

## 6.6 IMPLEMENTATION AND SYSTEM CONFIGURATIONS

All experiments were conducted using:

- **Hardware:** A laptop with NVIDIA RTX 4070 for MuJoCo simulations and forward predictions.
- **Language Models:** GPT-4o via OpenAI API;
- **Physics Simulators:** MuJoCo for environment modeling and trajectory evaluation.
- **Evaluation Interface:** A custom Python simulator for Tetris and real-time rendering with frame capture for trajectory visualization.

## 6.7 GRAPH MODELS

### 6.7.1 TRAINING OF DG-MOTION ATTENTION

**Training data generation.** We synthesize star-graphs with $n=6$ nodes (one master and five targets). Each node is assigned a random 3D position $\mathbf{x} \sim U(-10, 10)^3$, a random unit direction, and a speed $s \sim U(0.5, 1.0)$, yielding $\mathbf{v} = s\,\hat{\mathbf{v}}$. Half of the samples are labeled *collision*: we pick a random target and set its velocity to intercept the master's future position at horizon $t=3$s; the remaining half are *safe*. We compute a physically interpretable risk score at $t+\Delta t$ ($\Delta t=0.01$s) by combining (i) inverse distance, (ii) directional alignment (cosine), and (iii) speed via sigmoids with weights $(w_d, w_{dir}, w_v) = (0.34, 0.33, 0.33)$. Edges from the master to each target are labeled positive if risk $> \tau$ ($\tau=0.75$ by default). Node features are [isMaster, $\mathbf{x}_t$, $\mathbf{v}_t$] at $t$ and $t+\Delta t$, and edge attributes are relative displacements w.r.t. the master, yielding a star graph of $\mathcal{O}(n)$ edges per sample.

**DiffGraphormer (DG-Motion Attention).** Our model is a lightweight variant inspired by Graphormer (Ying et al., 2021), but implemented with TransformerConv (PyG) and explicit edge features. We encode nodes and edges with linear layers and apply a TransformerConv (with edge features) as the relational backbone. During the forward pass, differential motion $(\mathbf{x}_{t+\Delta t} - \mathbf{x}_t)/\Delta t$ provides velocity cues to construct edge attributes aligned with the data generator (distance, direction, speed). An edge head aggregates endpoints ($h_{ij}=h_i+h_j$) and outputs a sigmoid probability for each master→target edge. We train with binary cross-entropy; in deployment, we favor high-recall thresholds (e.g., >90% recall with ∼70% accuracy) to minimize missed hazards that would prevent APEX from triggering physics rollouts.

**Training setup.** We train edge-level hazard predictors on the synthetic star-graph dataset. We split data 80/20 for train/val and use a batch size of 1 (variable-size graphs), Adam (lr=$10^{-3}$), 100 epochs, and $\Delta t$=0.01s. Models include DiffGraphormer (TransformerConv with edge features) and ablations DiffGAT/DiffGCN.

**Loss & class balance.** We optimize binary cross-entropy with logits and a positive class weight $w^+=(1-\pi)/\pi$ computed from the dataset prior $\pi$ (positive ratio). Evaluation. We report edge-level accuracy and recall on the validation split with a 0.5 decision threshold, prioritizing high recall to avoid missed hazards that would bypass APEX's simulation trigger. Trained weights are saved for deployment.

### 6.7.2 DESIGN PRINCIPLE

Although our goal is not to benchmark graph architectures, one might ask why we place a graph module after the perception stack. This is an engineering choice. The graph plays two complementary filtering roles: (i) *interaction filtering:* in a cluttered scene, not all pairwise (or higher-order) interactions are task-relevant. Curating a sparse, task-conditioned interaction set prevents overlong contexts for LLM/VLM-based reasoning; and (ii) *temporal saliency filtering:* selecting only the most informative current frames (*triggers*) substantially reduces compute and relaxes the FPS requirements for downstream modules.

Beyond filtering, a scene graph offers a clean interface for switching between the physical world and natural language while retaining spatial structure and object state. Concretely, it preserves object-centric coordinates and attributes, implicitly maintaining an approximate SE(3) consistency that can be online updated.

The practical upside is that graph modeling is a mature area: from annotation pipelines to training recipes, we can leverage well-established methods rather than inventing bespoke machinery.

**Directions and Examples.** We highlight several graph-based avenues that align with our system:

- **Physical Interaction Graphs** (e.g., falling/moving dynamics): encode contact, support, and relative motion to gate physics queries and rollouts.

- **Semantic Hybrid Graphs**: integrate symbolic object categories with continuous physical states, enabling reasoning that links high-level semantics (e.g., "cup") with contextual properties (e.g., "full of water", "hot").

- **Safety Graphs**: augment nodes and edges with risk labels and constraints, supporting safety-aware planning and intervention (Huang et al., 2025).

- **Partial Complement Graphs**: expand partial observations (e.g., "a hand") into complete object groups (e.g., articulated human joints).

- **Spatio-Temporal Graphs**: capture objects whose motion patterns deviate from typical dynamics, such as those that suddenly appear or exhibit anomalous trajectories.

- **Counterfactual Graphs**: represent causal structures that support "what-if" reasoning (e.g., if object A had not collided with object B, would B still move?), enabling stronger generalization and interpretability.

## 6.8 Physical Engine / World Model

Simulation-based methods inevitably face both sim-to-real shift and partial observability. If we restrict the scope to Newtonian mechanics, information-theoretic considerations suggest that, given sufficiently rich observations of the real world, the Newtonian laws provide the most compact and faithful model. Under partial observation, the primary challenge is therefore accurate sensing and identification of the entities present in the scene, rather than entangling object categories (e.g., "apple," "cup") with specific motion patterns (e.g., free fall). Although one may train a model to approximate linear operators, and linearity is central to Newtonian mechanics, this introduces additional training cost and instability: we cannot guarantee that the model has internalized the gravitational constant or that such constants scale coherently across all motions.

Attempts to realize a purely learned *world model* that performs physical forward prediction with large sequence models (e.g., Transformers) inherit these issues (see Appendix 6.1). A lightweight, hybrid world model layered on top of a physics engine may be promising, but we leave a thorough exploration to future work.

**Limitations.** Beyond sim-to-real shift, partial observability, and the deliberate restriction to Newtonian regimes, both real and simulated environments exhibit chaotic dynamics. Measurement noise implies that long-horizon simulations accumulate bounded error. For physics engines, however, existing numerical analysis provides stability and error bounds, enabling principled confidence assessments. In contrast, black-box learned world models generally lack such calibrated uncertainty and verifiable error guarantees, which remains a key limitation.

## 6.9 Action Space Analysis

Assume an agent with $n$ degrees of freedom (DOF) and $k$-step rollouts. A naive complexity is:

$$O\big((n \cdot l)^k\big),$$

where $l$ denotes the discretization granularity.

In practice, we employ a coarse-to-fine search strategy: early steps use low-resolution discretization (e.g., $5°$), and the resolution is progressively refined near step $k-1$. Thanks to the Markov property, redundant rollouts are avoided by caching and pruning previously visited states.

Thus, the effective complexity becomes:

$$O\big( \min \big((n \cdot l_1)^k, \ s \cdot l_2\big)\big),$$

where $s$ is the number of reachable states, and $l_1, l_2$ denote coarse and fine resolutions, respectively.

Unlike Bellman-style methods, APEX avoids learning a high-dimensional value function, naturally supports heuristic pruning, and scales efficiently.

**Computational Overhead.** A common concern is the computational cost of simulation-based rollouts. In APEX, rollout simulates $n$ objects for $k$ seconds with step size $\Delta t$, resulting in:

$$O\left(\frac{n \cdot k}{\Delta t}\right) \text{ operations.}$$

For example, with $n = 100$, $k = 1$, and $\Delta t = 10^{-4}$, the rollout involves $10^4$ steps. A standard CPU core can handle approximately $10^9$ FLOPs/s, so this costs less than 1 ms runtime per rollout. Since simulations are fully parallelizable, APEX runs efficiently on CPUs without requiring specialized hardware.

The physics engine ensures both interpretability and real-time feasibility. Despite relying on high-fidelity simulations, APEX remains efficient and tractable.

Moreover, the combined complexity of action rollout and simulation is multiplicative. However, all simulations are independent, and each frame involves only linear-time physics computation per object. This structure naturally enables GPU-level parallelism.

As an illustration, consider a brute-force rollout of 100,000 actions over 1 second with timestep $\Delta t = 0.01$. Assuming 10 objects are active per frame, each requiring $\sim 10^3$ FLOPs, the total cost is:

$$(100{,}000 \text{ actions}) \times (100 \text{ frames}) \times (10 \text{ objects}) \times (10^3 \text{ FLOPs}) = 10^{11} \text{ operations.}$$

Such a workload is easily handled by modern GPUs and can be completed within milliseconds through parallel computation.

**Replacement.** The APEX rollout module is drop-in replaceable with pretrained RL policies or Monte Carlo Tree Search (MCTS) planners for candidate action generation; our caching/pruning and simulation interfaces remain unchanged and continue to provide the same complexity and parallelism benefits.

## 6.10 ERROR ANALYSIS FOR PHYSICS QA

Despite its general reasoning strength and applying of correct physical laws, GPT-4o often struggles with numerical accuracy especially on displacement calculations, which is crucial in real-world application like auto driving. We sampled failure cases across five representative tasks to understand the underlying issues.

**(1) 3D Linear Motion: Numerical Errors**

---

**Linear Motion Numerical Error**

**Question:** An object starts at $(0, 0, 0)$ with an initial velocity of $[13.15, 1.48, 6.23]$ m/s and an acceleration of $[-2.77, 2.36, 1.77]$ m/s. What are its velocity $vx, vy, vz$ and displacement $dx, dy, dz$ after 9.03 seconds?

**LLM Reasoning:** To find the velocity and displacement of the object after 9.03 seconds, we use the kinematic equations. The final velocity in each direction can be calculated using the formula $v = u + at$, where $u$ is the initial velocity, $a$ is the acceleration, and $t$ is the time. The displacement in each direction can be calculated using the formula $s = ut + 0.5at^2$.

For the x-direction: Initial velocity $(u_x)$ = 13.15 m/s, acceleration $(a_x)$ = -2.77 m/s². Final velocity $(v_x) = 13.15 + (-2.77) \times 9.03 = -11.8851$ m/s. Displacement $(d_x) = 13.15 \times 9.03 + 0.5 \times (-2.77) \times (9.03)^2 = \mathbf{59.364315}$ m.

For the y-direction: Initial velocity $(u_y)$ = 1.48 m/s, acceleration $(a_y)$ = 2.36 m/s². Final velocity $(v_y) = 1.48 + 2.36 \times 9.03 = 22.7708$ m/s. Displacement $(d_y) = 1.48 \times 9.03 + 0.5 \times 2.36 \times (9.03)^2 = \mathbf{114.94686}$ m.

For the z-direction: Initial velocity $(u_z)$ = 6.23 m/s, acceleration $(a_z)$ = 1.77 m/s². Final velocity $(v_z) = 6.23 + 1.77 \times 9.03 = 22.2101$ m/s. Displacement $(d_z) = 6.23 \times 9.03 + 0.5 \times 1.77 \times (9.03)^2 = \mathbf{147.014145}$ m.

**LLM Answer:**

```
{
    "velocity_x": "-11.8851",
    "velocity_y": "22.7708",
    "velocity_z": "22.2101",
    "displacement_x": "59.3643154", (wrong)
    "displacement_y": "114.94686",
    "displacement_z": "147.014145"
}
```

**Ground Truth:**

```
{
    "velocity_x": -11.86,
    "velocity_y": 22.79,
    "velocity_z": 22.21,
    "displacement_x": 5.81,
    "displacement_y": 109.58,
    "displacement_z": 128.42
}
```

**Error:** Miscalculation on `displacement`.

---

**(2) 3D Circular Motion: Numerical Errors**

### (2) 3D Circular Motion Numerical Error

**Question:**

- **Motion Type:** Circular Motion
- **Radius:** 2.49 meters
- **Speed:** 8.75 m/s
- **Angular velocity:** 3.51 rad/s
- **Time:** 5.95 s
- **Plane:** xy-plane
- **Task:** Compute its position $(x_B, y_B, z_B)$, assuming it starts at $(r, 0, 0)$.

**LLM Reasoning:**
To find the position of the object in circular motion, we use the parametric equations for circular motion. The object starts at position $(r, 0, 0)$ and rotates in the xy-plane. Its position at any time $t$ is given by:

$$(x_B, y_B) = (r \cdot \cos(\theta), r \cdot \sin(\theta))$$

where $\theta$ is the angular displacement. The angular displacement $\theta$ is calculated as:

$$\theta = \omega \times t = 3.51 \times 5.95 = 20.8845 \text{ rad}$$

Thus:

$$x_B = 2.49 \cdot \cos(20.8845) \quad \text{and} \quad y_B = 2.49 \cdot \sin(20.8845)$$

Since the motion is in the xy-plane, $z_B = 0$.

**LLM Answer:**

```
{
    "x_B": "0.57",
    "y_B": "-0.51",
    "z_B": "0"
}
```

**Ground Truth:**

```
{
    "x_B": 0.3206,
    "y_B": -0.7001,
    "z_B": 0.0
}
```

**Error:** Miscalculated both $x_B$ and $y_B$. Likely due to incorrect theta or trig eval.

### (3) 3D Projectile Motion: Axis Misinterpretation

### (3) 3D Projectile Motion: Axis Misinterpretation

**Question:**
A projectile is launched from $(0, 0, 0)$ with an initial velocity of $[7.68, 18.83, 11.81]$ m/s from x-y plane. Calculate its flight time, maximum height $(h)$, and range $(dx, dy, dz)$.

**LLM Reasoning:**
To solve the problem, we break down the motion into its components. The initial velocity vector is given as $[7.68, 18.83, 11.81]$ m/s, where 7.68 m/s is the x-component, 18.83 m/s is the y-component, and 11.81 m/s is the z-component. The projectile motion is influenced by gravity acting in the negative y-direction with an acceleration of $-9.81$ m/s. The flight

time can be found by determining when the projectile returns to the $y = 0$ plane. Using the kinematic equation for vertical motion $y = v_0 y * t + 0.5 * a * t^2$, we set $y = 0$ and solve for $t$. The maximum height is achieved when the vertical velocity is zero, calculated using $v = v_0 y + a * t$. The range in each direction is calculated using the horizontal components of motion, which are not affected by gravity. The horizontal ranges $(dx, dz)$ are calculated as $v_0 x * t$ and $v_0 z * t$, respectively. The maximum height is calculated using the vertical motion equation with the time at maximum height.

**LLM Answer:**

```
{
  "flight_time": "3.84",
  "maximum_height": "18.06",
  "range_x": "29.47",
  "range_y": "0",
  "range_z": "45.32"
}
```

**Ground Truth:**

```
{
  "flight_time": 2.41,
  "maximum_height": 7.11,
  "range_x": 18.49,
  "range_y": 45.34,
  "range_z": any.
}
```

**Error:** The model confuses the vertical axis with the forward direction, mistakenly assigning horizontal motion to the $z$-axis and ignoring vertical displacement on $y$.

**(4) 3D Multi-Object Motion: Complex Reasoning with Mixed Physics Types**

### 3D Multi-Object Motion Error (LLM Inaccuracy in Hybrid Settings)

**Question:**
We have three objects moving in different types of motion in 3D space:

1. **Object A - Linear Motion**
   - Initial Velocity: $[11.41, 1.01, 6.45]\,m/s$
   - Acceleration: $[-0.7, 0.8, 1.55]\,m/s^2$
   - Time: $5.18\,s$
   - Compute the final position $(x_A, y_A, z_A)$, assuming it starts at $(0, 0, 0)$.

2. **Object B - Circular Motion**
   - Radius: $2.49\,m$
   - Angular Velocity: $1.17\,rad/s$
   - Time: $5.26\,s$
   - Rotating in the xy-plane.
   - Compute the position $(x_B, y_B, z_B)$, assuming it starts at $(r_B, 0, 0)$.

3. **Object C - Projectile Motion**
   - Initial Speed: $[21.47, 12.09, 21.6]\,m/s$
   - Time: $6.57\,s$
   - Compute the position $(x_C, y_C, z_C)$, assuming it starts from $(0, 0, 0)$ in the xy-plane.

**Reasoning:**

- For **Object A (Linear Motion)**, we use the kinematic equation for linear motion:

$$\text{Position} = \text{Initial Velocity} \times t + 0.5 \times \text{Acceleration} \times t^2$$

- For **Object B (Circular Motion)**, the position is calculated using the parametric equations:

$$(x_B, y_B) = (r \cdot \cos(\omega \cdot t), r \cdot \sin(\omega \cdot t)), \quad z_B = 0$$

- For **Object C (Projectile Motion)**, the position is calculated for each component as:

$$x_C = v_x \times t$$
$$y_C = v_y \times t$$
$$z_C = v_z \times t - 0.5 \times g \times t^2$$

where $g$ is the gravitational acceleration.

**LLM Answer:**

```
{
    'pos_A': {'x_A': '51.698247', 'y_A': '8.06772', 'z_A':
    '48.568195'},
    'pos_B': {'x_B': '1.176498', 'y_B': '2.140898',
    'z_B': '0'},
    'pos_C': {'x_C': '141.0939', 'y_C': '79.46213',
    'z_C': '48.842055'}
}
```

**Ground Truth:**

```
{
    'pos_A': {'x_A': 49.71, 'y_A': 15.96, 'z_A': 54.21},
    'pos_B': {'x_B': 2.47, 'y_B': -0.32, 'z_B': 0},
    'pos_C': {'x_C': 141.06, 'y_C': 79.43, 'z_C': any}
}
```

**Error:** The questions are too complex, and the reasoning parts are correct, but the calculations are wrong.

## (5) 3D Collision: Fails to Predict Future Collision

### (5) 3D Collision: Fails to Predict Future Collision

**Question:**
Two objects with masses $8.4\,kg$ and $5.91\,kg$ are located at positions:

$$\text{Object 1 Position: } [-2.8817, -0.4259, -2.0413]$$
$$\text{Object 2 Position: } [-0.4778, 1.7133, 0.4933]$$

The radius of both spheres is $0.5$.
They are moving with velocities:

$$\text{Object 1 Velocity: } [1.999, 1.779, 2.108]\,m/s$$
$$\text{Object 2 Velocity: } [-1.273, -1.133, -1.342]\,m/s$$

Assuming an **elastic collision**, determine:

- Will they collide?
- If they collide, what are their final velocities?

**LLM Reasoning:**

To determine if the two objects will collide, we first calculate the distance between their centers using the Euclidean distance formula:

$$d = \sqrt{(x_2 - x_1)^2 + (y_2 - y_1)^2 + (z_2 - z_1)^2}$$

If the distance $d$ is less than or equal to the sum of their radii (1.0), they are already colliding. If not, we proceed to check their relative velocities to determine whether they are moving towards each other.

The relative velocity vector $\mathbf{v_{rel}}$ is given by:

$$\mathbf{v_{rel}} = \mathbf{v_2} - \mathbf{v_1}$$

We then compute the dot product of $\mathbf{v_{rel}}$ and the vector connecting the centers. If the dot product is **negative**, it indicates that the objects are moving towards each other.

In this case:
- The initial distance is approximately $4.558$, which is greater than the sum of the radii 1.0. - The dot product of the relative velocity vector and the center-to-center vector is **negative**, indicating that the objects will collide.

For an **elastic collision**, we apply the conservation of momentum and kinetic energy to calculate the final velocities of both objects.

**LLM Answer:**

```
{
   "will_collide": "true",
   "velocity_1":
       {"vel_1_x": "-1.2731450866327967",
        "vel_1_y": "-1.1329265996055307",
        "vel_1_z": "-1.3423728509665511"},
   "velocity_2":
       {"vel_2_x": "1.9989536980325082",
        "vel_2_y": "1.7787979073700424",
        "vel_2_z": "2.107647590798081"}
}
```

**Ground Truth:**

```
{
   "will_collide": "true",
   "velocity_1":
   {"vel_1_x": -0.7, "vel_1_y": -0.63, "vel_1_z":  -0.74},
   "velocity_2":
   {"vel_2_x":  2.57, "vel_2_y":  2.29, "vel_2_z":  2.71}
}
```

**Error:**

Although LLM correctly identified the two object will collide, it failed to calculate the velocities by Newton's third law of motion but just swap them.

## 6.11 CASE STUDIES: TETRIS PLANNING

We analyze qualitative behavioral differences across four models on identical Tetris configurations, with visualizations shown in Figures 7–10. Each model was given the same initial board states and action budget.

These case studies highlight the crucial role of physics-based foresight in long-horizon spatial planning. APEX not only reacts to the present state, but also reasons about the physical impact of future placements, resulting in more strategic and compact gameplay.

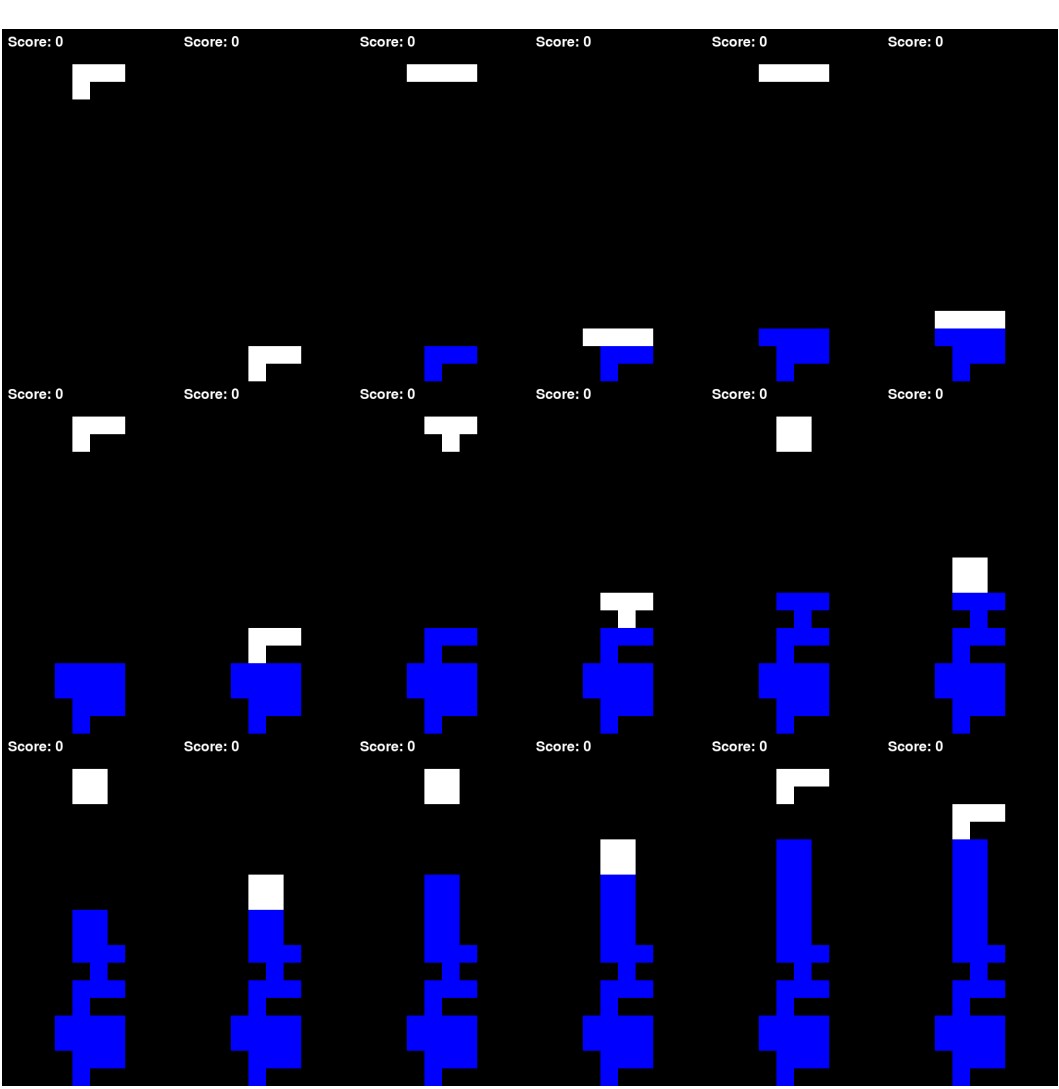

Figure 7: Performance of GPT-4o-mini. Despite various prompt engineering attempts, GPT-4o-mini consistently defaulted to the `down` action regardless of board state. As a result, the pieces were dropped directly without any lateral movement or rotation, quickly leading to high towers and early termination. The model lacks basic spatial foresight and cannot anticipate block alignment or stability.

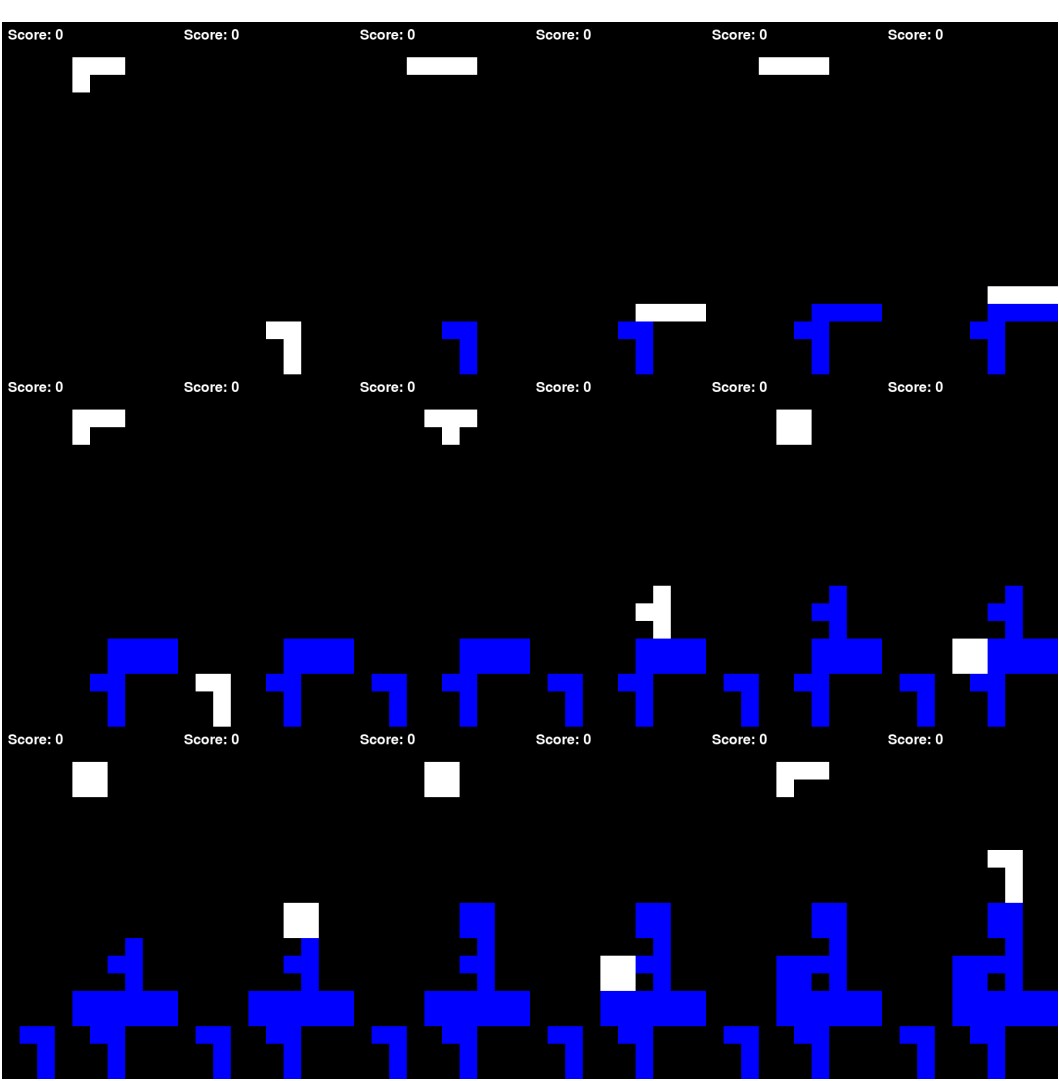

Figure 8: Performance of GPT-4o. The full GPT-4o model shows slight improvement over its miniature counterpart by occasionally moving blocks laterally. However, it frequently misjudges horizontal distances and fails to align pieces with open gaps. This often results in suboptimal placements and growing bumpiness. The model demonstrates some reactive planning but lacks consistent spatial optimization.

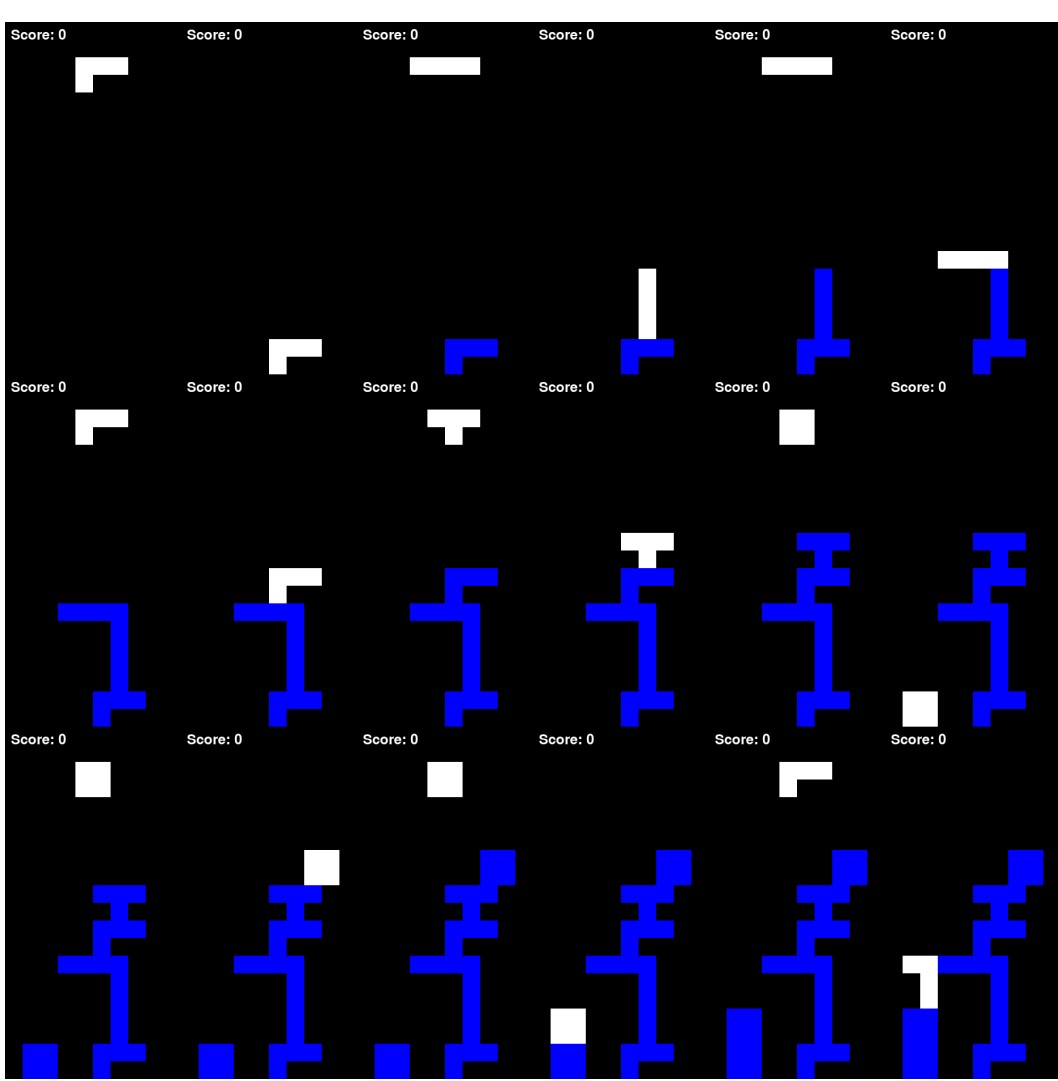

Figure 9: Performance of VLM. Incorporating visual perception enables better state awareness, but the VLM model exhibits a strong reluctance to rotate pieces. For instance, long vertical bars are often dropped in upright orientation at the center of the board, creating tall columns that destabilize subsequent placements. The inability to rotate blocks limits the model's flexibility and leads to inefficient spatial usage.

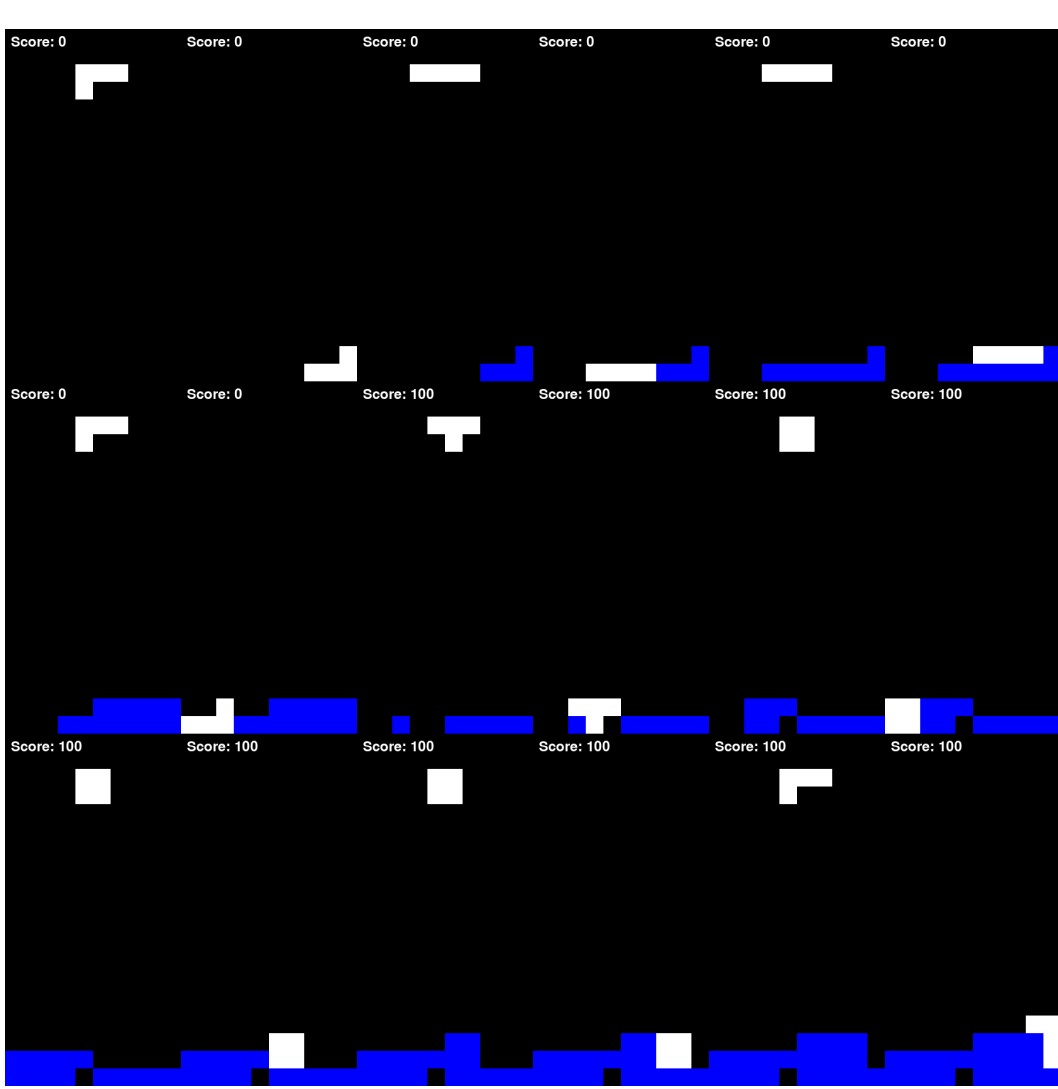

Figure 10: Performance of APEX. In contrast, APEX consistently produces compact, flat stack configurations with minimal bumpiness and high spatial efficiency. It is the only model capable of clearing lines and accumulating positive scores. By simulating multi-step outcomes using a physics engine and selecting actions based on predicted results, APEX avoids naive placement and optimizes long-term board stability. Notably, the resulting stack heights are approximately one-fourth that of the baseline models, clearly demonstrating the benefits of anticipatory physical reasoning.

## 6.12 EXAMPLES FROM DYNAMIC OBSTACLE AVOIDANCE

Figures 11–15 contrasts navigation behaviors under four conditions, highlighting the role of APEX in guiding physically informed decision-making in dynamic obstacle environments. In Table 6, we found that GPT-4o-mini tends to exploit a shortcut in decision-making: it selects any path labeled as "Safe" without considering the actual distance to the obstacle. Specifically, in the final timestep of one scenario, we labeled a move as "Safe" if the distance to the nearest obstacle exceeded a threshold of 0.5 meters. One such option (moving left) had a distance of 0.54 meters, just above the threshold, while alternative paths offered significantly safer margins (over 2.0 meters). GPT-4o-mini simply selected the first available "Safe" option without comparison.

GPT-4o occasionally made similar mistakes when not explicitly prompted with instructions such as "choose the path farthest from the obstacle." However, GPT-4o-mini consistently followed this suboptimal policy, defaulting to a static heuristic.

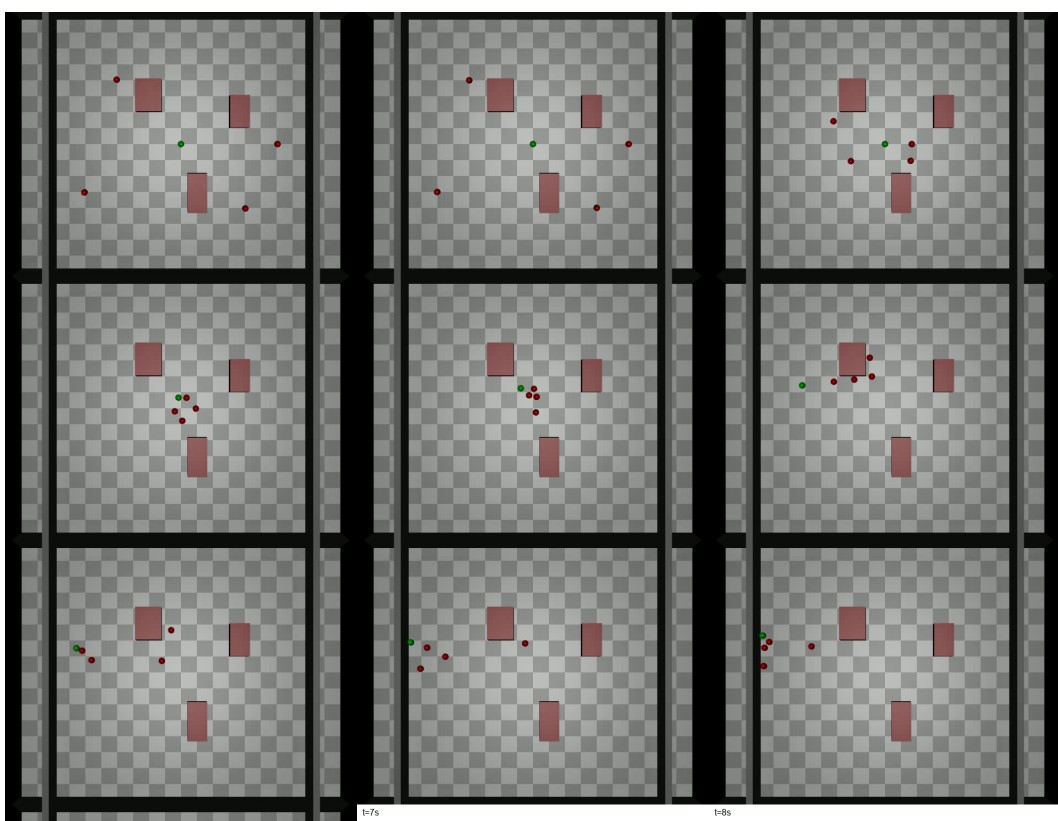

Figure 11: Performance of GPT-4o-mini. GPT-4o-mini fails to react altogether, remaining static even as a moving obstacle approaches. This indicates a lack of temporal prediction or awareness of imminent collision.

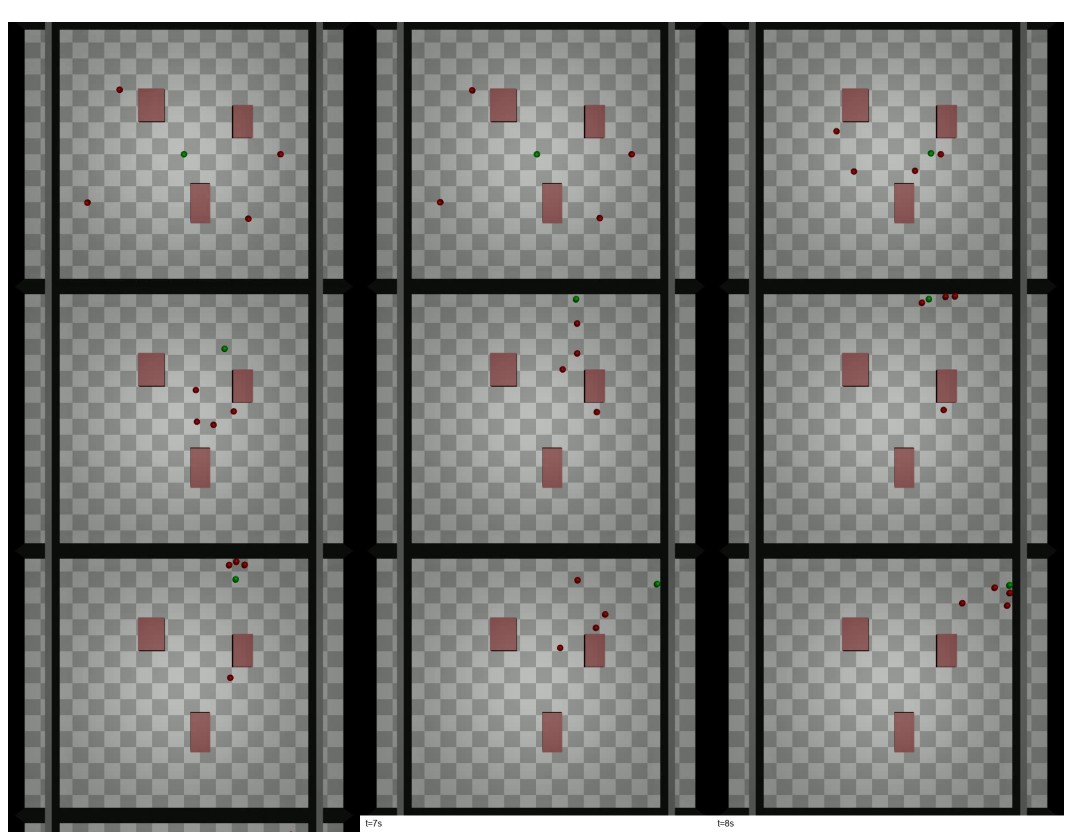

Figure 12: Performance of GPT-4o. GPT-4o recognizes the presence of a moving object but selects an incorrect evasive direction, resulting in a direct collision. While perceptual awareness is present, the absence of predictive modeling leads to poor decision quality.

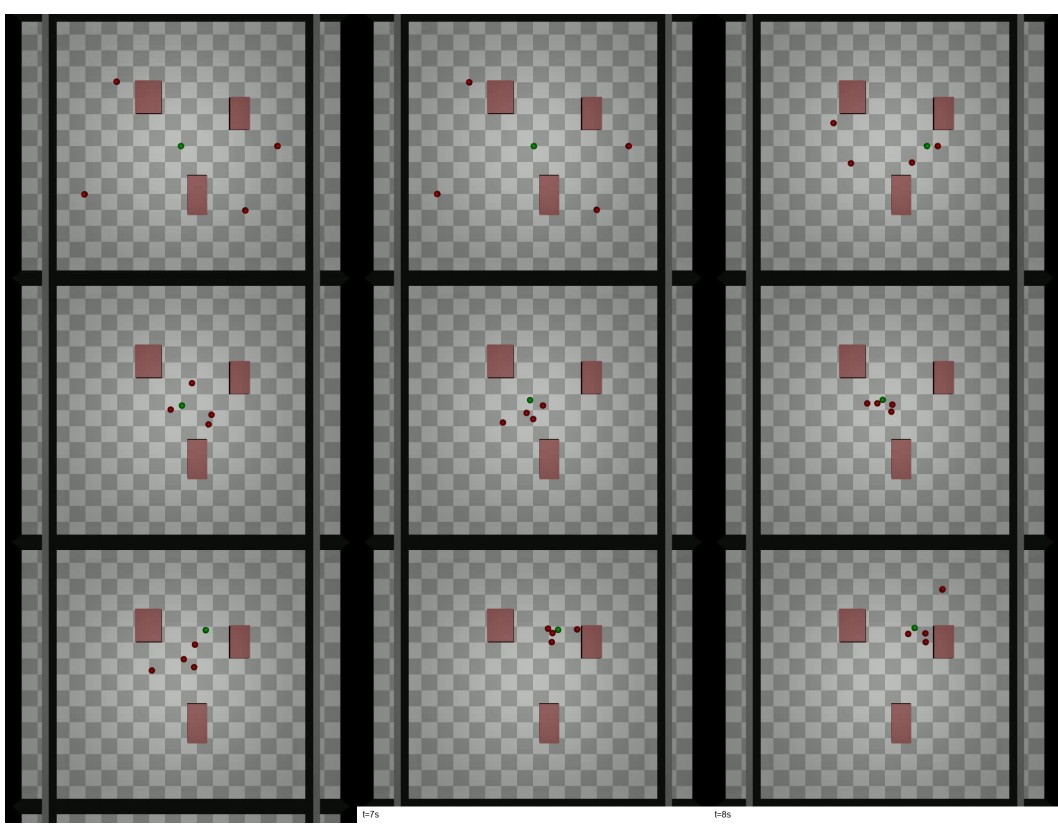

Figure 13: Performance of VLM. We observe consistent failure patterns in the VLM-based baseline across multiple scenarios. In some cases, the model produces responses such as *"Sorry, I can't help with that"*, indicating that it is unable to generate actionable plans when faced with ambiguous or dynamic input. More critically, the model often misjudges object displacement or relative movement, leading to physically invalid plans, such as walking into obstacles that are visibly approaching. This suggests a lack of grounded numerical estimation and forward reasoning capability, which are necessary for real-world spatial tasks.

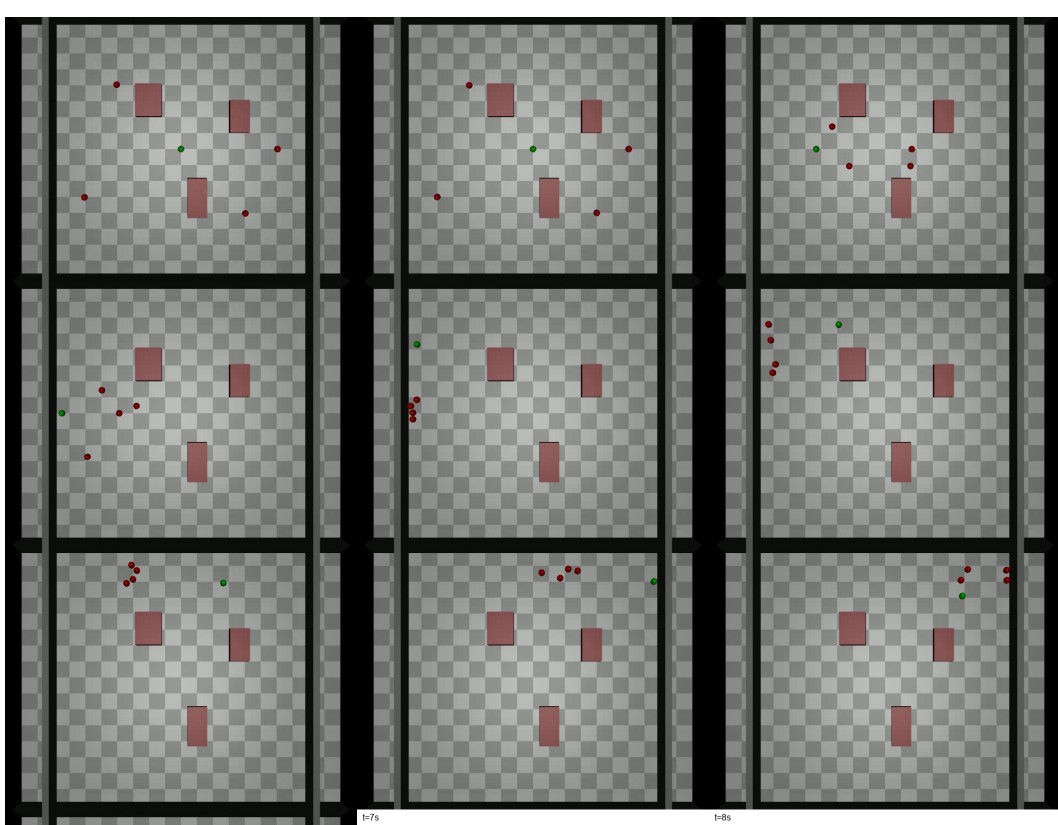

Figure 14: Performance of APEX(GPT-4o-mini). GPT-4o-mini operates under APEX guidance but occasionally disregards simulated risk evaluations, choosing paths that minimize immediate distance to the goal, ironically aligning with the obstacle's trajectory. This suggests limited integration of long-term consequence awareness.

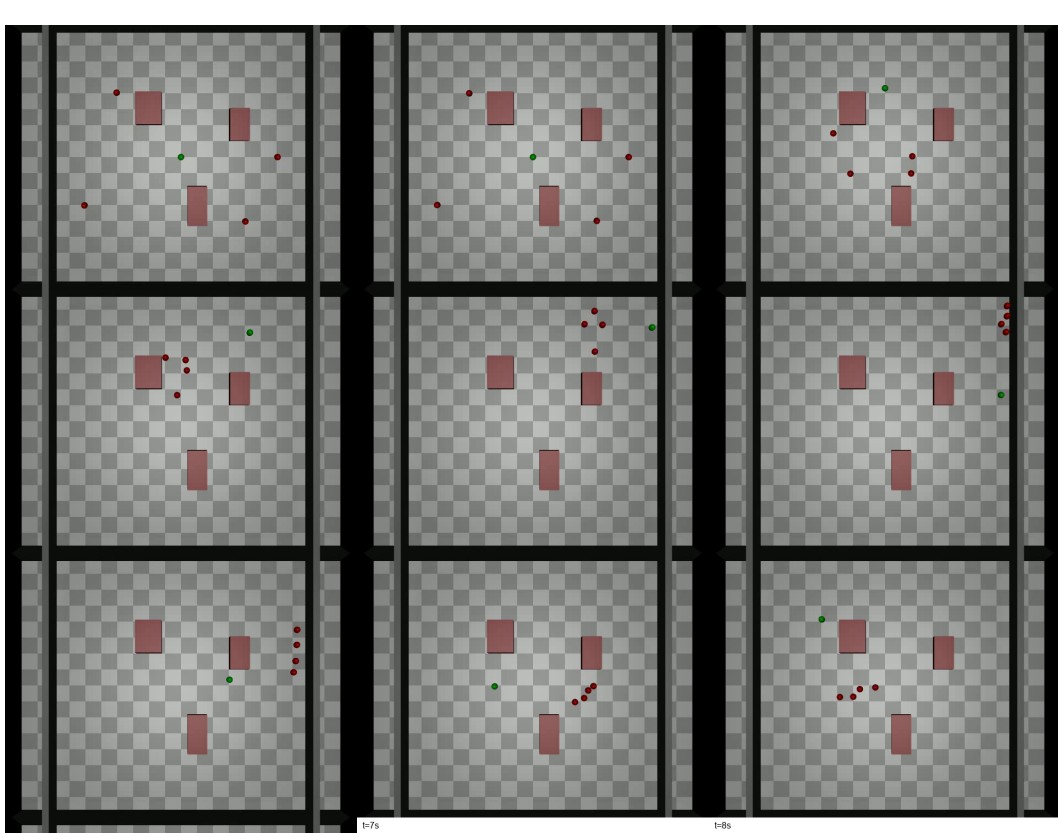

Figure 15: Performance of APEX(GPT-4o). GPT-4o, with full APEX support, exhibits anticipatory behavior, dynamically adjusting its trajectory to avoid the obstacle while maintaining movement toward the goal. Notably, the agent even "orbits" around the obstacle when direct paths are unsafe, demonstrating flexible foresight and real-time risk mitigation.

