# OpenReview forum: "APEX: Empowering LLMs with Physics-Based Task Planning for Real-time Insight"
_ICLR.cc/2026/Conference — ICLR 2026 Conference Withdrawn Submission_

### Official Review · Reviewer_LBxQ · 2025-10-15

**Soundness:** 2
**Presentation:** 3
**Contribution:** 2
**Rating:** 4
**Confidence:** 4

**Summary:**

This paper introduces APEX, a framework that enhances LLMs with physics-based reasoning capabilities for real-time task planning in dynamic environments. APEX constructs structured interaction graphs from environmental snapshots, uses a Graphormer-based attention mechanism to identify salient dynamic interactions, and performs forward simulations via a MuJoCo to predict action outcomes. The LLM then selects optimal actions based on these simulations. The authors evaluate APEX on three benchmarks: a Physics Reasoning Benchmark, Tetris for long-horizon planning, and Dynamic Obstacle Avoidance. Results show that APEX significantly outperforms baseline LLMs and VLM-based models, demonstrating improved physical reasoning, foresight, and real-time adaptability.

**Strengths:**

1. Introduces a novel graph-simulation-loop architecture to help LLMs in physical reasoning.

2. Evaluation across three diverse benchmarks with strong baselines and ablations.

3. Well-organized and accessible, with clear explanations of both high-level ideas and implementation details.

**Weaknesses:**

1. The framework's applicability is currently limited to motion-related problems, particularly collision prediction. This narrow scope and its reliance on a specific simulator make it brittle; adapting it to new tasks or physical phenomena likely requires significant re-implementation.
2. To demonstrate generalizability, future work should include experiments in more complex settings, such as those with cluttered scenes or deformable objects.
3. Although APEX's modularity is a strength, the paper provides limited analysis of the design choices involved, such as the trade-offs between different graph encoders or simulators. Several architectural questions remain: for instance, are the graph and simulation modules strictly decoupled, or could they be jointly fine-tuned with the LLM? Furthermore, have the authors considered integrating learned world models (e.g., neural physics engines) as potential replacements for MuJoCo?
4. The real-world experiments demonstrate the system's capability under controlled conditions. However, it is unclear how sensitive APEX is to practical challenges like perception noise or delays in the dynamic graph construction process.

**Questions:**

See weaknesses. I am willing to discuss and improve the paper.

---

> ### Author Response · Authors · 2025-11-22
>
> W4.1:
> **“Applicability is limited to motion-related collision prediction.”**
> **A:** We respectfully disagree on the applicability of our framework..
> The real-world third-party collision intervention experiment already illustrates that, *without any explicit prior prompt*, the LLM–physics loop can reason about rare safety-critical causal structures such as
> **“child → car → collision → unacceptable → intervene.”**
>
> This type of semantic–physical reasoning generalizes naturally to many real-world embodied AI domains, e.g.:
>
> - An excavator halting operation when a person enters an unsafe zone.
> - Identifying unsafe or infeasible actions due to weight, friction, or instability.
>
> With an appropriate graph model, a household robot can detect a boiling kettle that may trigger a fire hazard and with a LLM to decide to turn it off or notify humans.
>
> The current paper focuses on *motion-level validation* because it is an important topic, but the architecture itself is not tied to motion-only tasks.
>
>
>
>
>
>
>
> ---
>
> W4.2:
> **“Cluttered scenes and deformable objects.”**
> **A:** We agree that both cluttered scenes and deformable / liquid objects are important and challenging directions. Our current experiments focus on rigid-body motion because it allows us to isolate the effect of physics-grounded planning under well-understood dynamics. However, the framework itself is not restricted to this regime: the physics module can, in principle, be instantiated by a soft-body or fluid simulator, or by a learned neural physics model, as long as it exposes the same interface (state rollout under candidate actions). For cluttered scenes, the graph module is particularly suitable for grouping and saliency filtering over many objects (e.g., clustering irrelevant background items vs. agents and obstacles), which we view as a natural extension rather than a fundamental limitation of the current design.
>
> ---
> W4.3:
> **“Trade-offs between graph encoders, simulators, and architectural choices.”**
> **A:** We appreciate this suggestion.
> We will include a more detailed discussion of these trade-offs in the revision.
>
> ---
> W4.4:
> **“Sensitivity to perception noise and module latency.”**
> **A:** We address both issues empirically:
>
> - In PHYRE, we intentionally injected noise and observed improved robustness using relational filtering.
> - In the real-world robotic experiment, we applied smoothing across frame windows to handle perception noise.
>
> In practice, the **LLM is the dominant latency source** (seconds), while perception, graph construction, and simulation are millisecond-level.
> This is exactly why the graph module filters salient interactions:  to reduce how often the LLM must be invoked in the planning loop.
>
> ---
>
> ### Discussion on Trainability and Modularity
>
> Although we did not implement the full PGLPA system in this submission, the design explicitly supports **online updating and learning**:
>
> - **Updating physical priors.**
>   Object properties such as mass or friction can be updated during interaction.
>   This is one reason we did not adopt purely vision-based world models, but if there is one light-wise structured world model where we can online update the physical parameters, we are willing to test it.
>
> - **Learning across modules.**
>   Success/failure of simulated or real executions can update action models, graph encoders, memory modules, and can even fine-tune LLM spatial reasoning.
>   Because the simulator is SE(3)-consistent, the system can auto-label symbolic spatial relations (directions, contact forces, geometry), enabling richer grounding on world knowledge of large models.

---

### Official Review · Reviewer_ToiG · 2025-10-17

**Soundness:** 1
**Presentation:** 2
**Contribution:** 2
**Rating:** 2
**Confidence:** 4

**Summary:**

This paper proposes APEX, a framework intended to augment LLMs with explicit physics-based foresight for task planning. It introduces a Perception–Graph–Language–Physics–Action pipeline that integrates a graph attention module and a physics simulator to provide physical rollouts as feedback to an LLM during decision making. The authors evaluate APEX on three benchmark domains—synthetic physics QA, Tetris planning, and dynamic obstacle avoidance—claiming substantial improvements over vanilla GPT-4o and VLM baselines.

**Strengths:**

1. Interesting high-level motivation. Bridging symbolic reasoning in LLMs with physically grounded modeling is an important and timely goal.
2. Attempt to unify physics reasoning and LLM planning. The modular architecture (graph → simulator → LLM → action) provides a readable system outline.

**Weaknesses:**

1. Conceptual novelty is limited. The core idea—using a physics engine to simulate candidate actions and feeding the results back to an LLM—is conceptually straightforward and has appeared in prior “simulation-in-the-loop” or “world-model prompting” works (e.g., Mind’s Eye, PiLoT, PhysVLM). The proposed Perception–Graph–Language–Physics–Action paradigm is mostly a re-labeling of existing perception-simulation-planning loops in robotics; there is no theoretical or algorithmic advance beyond modular composition.
2. Questionable experimental design and fairness. Benchmarks are non-standard. The “Physics Reasoning Benchmark,” “Tetris,” and “Dynamic Obstacle Avoidance” are all custom setups with unclear data availability or reproducibility.
3. Paper is not well written. The main text could not fully present the results and the analysis. I suggest putting some result figures from appendix to the main text. Table format should be unified and the figures need further improvement for clarity.

**Questions:**

See above.

---

> ### Author Response · Authors · 2025-11-22
>
> W3.1:
> **“Conceptual novelty is limited.”**
>
>
> **A:** We clarify here how our approach differs from prior simulation-in-the-loop methods such as Mind’s Eye [1] and PhysVLM [2].
>  Those works typically focus on one-shot static reasoning (e.g., predicting which ball lands first) and do not embed physics-forward prediction into a closed action-planning loop in SE(3).
>  In contrast, our work targets general task planning in dynamic environments, where the agent must reason about evolving states, possible interventions, and the feasibility of multi-step action sequences.
>  Moreover, we propose an explicit unified Perception → Graph → Physics → Language → Action framework, where:
> Perception provides structured object states;
>
>
> A graph module filters interaction saliency;
>
>
> Physics rollouts serve as a quantitative oracle;
>
>
> An LLM performs high-level reasoning and plan selection;
>
>
> Actions are grounded back into SE(3).
>
>
> To our knowledge, this explicit modular composition, with physics rollouts as a first-class component inside the LLM planning loop, has not been formalized in prior work.
>
> ---
> W3.2:
>
> **“Benchmarks are non-standard.”**
> **A:** We acknowledge that Physics QA, Tetris, and Dynamic Obstacle Avoidance are custom demonstrations.
> This is intentional:
> the purpose of these experiments is **not** to compete on existing benchmarks,
> but to isolate and illustrate two claims of our framework:
>
> 1. **Physics rollouts improve LLM reasoning**, and
> 2. **Predicted future states improve action feasibility evaluation.**
>
> To address concerns about reproducibility and external verification, we will additionally evaluate our method on the LIBERO-Pro [3] benchmark, as well as two more realistic and safety-critical settings:
>
> - **Third-party collision intervention**
> - **Over-heavy box pickup (infeasible-action detection)**
>
> These tasks highlight aspects that task-specific solvers cannot handle and that physics-grounded general planning is specifically designed for.
>
> We believe these added results will help clarify the framework’s strengths.
>
> ---
> W3.3:
> **“Paper writing, clarity, and figures.”**
> **A:** We appreciate these comments.
> We will restructure Section 4 to move key figures and ablations from the appendix into the main text, unify table formats, and improve figure clarity.
>
>
>
>
> [1] Liu, Ruibo, et al. "Mind's eye: Grounded language model reasoning through simulation." arXiv preprint arXiv:2210.05359 (2022).
>
> [2] Zhou, Weijie, et al. "Physvlm: Enabling visual language models to understand robotic physical reachability." Proceedings of the Computer Vision and Pattern Recognition Conference. 2025.
>
> [3] Zhou, Xueyang, et al. "LIBERO-PRO: Towards Robust and Fair Evaluation of Vision-Language-Action Models Beyond Memorization." arXiv preprint arXiv:2510.03827 (2025).

---

### Official Review · Reviewer_Z9SE · 2025-11-01

**Soundness:** 1
**Presentation:** 2
**Contribution:** 1
**Rating:** 2
**Confidence:** 4

**Summary:**

This paper introduces APEX, a framework for enhancing LLM’s physical reasoning capabilities for real-time task planning by integrating with simulations of physical engines. Given some prompt, the system is able to generate qualitative and quantitative predictions of the scene in text. Specifically, given two frames, APEX first creates a scene graph that captures object relations and interactions, then passes the current relational state to a physics engine to simulate possible futures under different actions. The resulting information is incorporated into a unified prompt for the LLM, which then selects the optimal action.

The paper introduces three benchmarks and evaluates APEX-enhanced LLMs with vanilla LLMs, and shows that APEX-enhanced LLMs outperforms vanilla LLMs on physics reasoning, Tetris, and obstacle avoidance.

**Strengths:**

- The paper attempts to address the physical reasoning limitations of LLMs by making use of physics engines.
- The paper provides some concrete examples of tasks and model outputs in the appendix.

**Weaknesses:**

- Overall, the introduced pipeline is very limiting. The framework makes many assumptions, but none are explicitly described in the paper. For example, the scene graph formulation assumes the scene contains mostly distinctive and rigid objects. Since the predictions are simulated with a physical engine, the method also assumes those objects are compatible with the simulator. What constraints are imposed on the acceptable objects in the scene? Moreover, the decision-making procedure requires a finite, enumerable set of actions, as the algorithm simulates each action one by one. This is largely infeasible in real decision-making tasks.
- The paper lacks baselines. It motivates the work by describing limitations of latent world models, RL, etc., but only compares APEX-enhanced LLMs with vanilla LLMs. What about latent world models, RL, or other methods as baselines? If they are not compatible, why motivate the paper based on their limitations?
- There is no description of how the physical engine is designed or used. How is the simulated scene created based on the current relation state? The paper only mentions MuJoCo in Figure 2, but what are the assumptions on the scene to make it simulatable by MuJoCo?
- The proposed pipeline appears to bottleneck information through text. If the system already has access to input frames, created scene graphs, and a physics engine, why is text used as the medium for reasoning and decision-making? Wouldn’t this cause loss of critical information, and isn’t it the case that not all scenes can be described accurately through text?
- The paper’s writing could be improved. It presents irrelevant information while missing some critical details about the method. E.g. Some works in the related work section appear misplaced,  for instance, it is unclear why RL is discussed, or why several non-RL papers are included in Section 2.3. Similarly, it is not clear why R3M is categorized under world models in Section 2.2.
- Some of the task figures should be moved to the main body. Currently, it is difficult to ground the evaluations, even though the paper provides some visuals in the appendix.

**Questions:**

- What are the assumptions on the scene, dynamics, and action space?
- The method only uses $G_t$ and $G_{t + \Delta t}$ as context. How does this capture second-order dynamics information such as acceleration? If such information must be provided through text, this imposes an additional requirement on the system that these values be known, yet information such as object velocity or acceleration is typically unavailable in real-world decision-making.
- Why use scene graphs? Although Appendix 6.7.2 provides some motivation in terms of interaction and temporal saliency filtering, these considerations mainly arise from converting everything into text. Why is this representation preferable to, for example, a latent world model, where such filtering would not be necessary?
- In the Tetris experiment, only APEX and vanilla LLMs are evaluated. What would the oracle performance be, e.g. from a classical planning algorithm or human players?
- While the paper provides some prompt and task examples in the appendix, what would a complete input–output pipeline look like for a specific task (e.g., Tetris)? What are the exact inputs and outputs of the Graphormer encoder and the physics engine? How are the entities represented and instantiated within the physics engine?

---

> ### Author Response · Authors · 2025-11-22
>
> W2.1:
> **Limitation of pipeline: dependence on simulator and discussion on action space**:
> **A: ** The method is not tied to any specific simulator. In our real-world experiment—where modern MuJoCo could not be deployed on a legacy robotic system—we manually implemented an Euler-forward update to predict future object states. This indicates that the pipeline only requires access to a forward-dynamics function on structured physical states, which is precisely the core principle underlying most modern simulators.
>
> Regarding the action space, our framework does not rely on exhaustive enumeration. In ongoing experiments, we replace the discrete action set with an LLM-driven planning loop expressed in a domain-specific language (DSL) linked to a skill library. These skills can be grounded either through the simulator or directly through the robot’s low-level actuators, demonstrating that the action module is fully modular and not restricted to the simplified version presented in the initial experiments.
>
> ---
> W2.2:
> **“Baselines.”**
> **A:** The paper motivates the work by discussing limitations of latent world models, RL, etc., and we currently compare APEX-enhanced LLMs with vanilla LLMs.
>  Our focus in this submission is to answer the question:
>  **“Does physics-grounded rollout, when placed inside the LLM planning loop, systematically improve reasoning and action evaluation?”**
>  To more directly address the baseline concern, we are in the process of adding comparisons against a generalist RL / VLA-style policy where possible, under the same observation and action interfaces.
>
> ---
> W2.3:
> **“How is the environment converted into a physical engine input?”**
> **A:** In the experiments on the main pages, perception outputs were directly obtained from the simulator.
> In the real-world experiment (Appendix 6.4), perception comes from RGB sensing on color segmentation.
> Importantly, the simulation module does **not** depend on the graph output; it operates directly on the structured perception outputs (object pose, geometry, estimated friction, etc.). The graph module is used solely for interaction saliency filtering.
>
> ---
>
>
> **“Rigid-body assumption.”**
> **A:**  We acknowledge that our current experiments operate under rigid-body dynamics, as this setting provides well-defined physical parameters and allows us to isolate the contribution of physics-grounded planning. Importantly, this assumption is not inherent to the framework. The physics module is designed as a replaceable “oracle”: any soft-body, deformable-object, or fluid simulator—and likewise any learned neural physics model—can be substituted as long as it provides forward rollouts under candidate actions. In this sense, the planning loop remains unchanged; only the physical predictor is swapped. Thus, the rigid-body restriction reflects an experimental choice rather than a structural limitation of the framework.
>
>
>
>
> ---
> W2.4:
> **“Why use text in the reasoning loop? Does this bottleneck information?”**
> **A:** Our long-term aim is a *general embodied AI framework*, not a task-specific controller.
> Text reasoning is needed because:
>
> 1. **High-level reasoning.**
>    Many tasks (e.g., third-party accident intervention in our real-world example) require semantic causal reasoning such as
>    *“child → car → collision → unacceptable → need intervention.”*
> Such reasoning is difficult to express or robustly learn through low-level RL or purely vector-valued policies.
> In particular, RL cannot be reliably trained for rare, safety-critical situations that may never occur in training data.
>
> 2. **Human–robot interaction.**
>    Language is the natural interface for general instructions.
>
> To mitigate information loss, the physics engine and perception modules operate on full continuous states; only a summarized, task-relevant subset is converted to text for the LLM. This is where the graph module helps: it focuses the description on salient interactions and relations, rather than transmitting raw pixel-level or full-state information.
>
> We adopt a Perception–Graph–LLM architecture rather than a fully end-to-end VLM because it allows **explicit control over the LLM’s context window of relations (e.g. collision and hazard)**, and enables **a smaller training scale**.
>
> ---
> W2.5 & W2.6:
> **“Writing of Section 2 (current methods).”**
> **A:** We thank the reviewer for pointing this out; we will revise and reorganize this section.
>
> ---
> **“Writing of Paper”**
> **A:** Our view is that the three main experiments in the paper are consistent and sufficient to demonstrate the core claim:
> ** “a physics- or simulator-enhanced LLM can operate within a closed-loop embodied system in dynamic environments.”**
> Additional components such as PHYRE, the real-world experiment, and extended baselines were moved to the Appendix to keep the main paper focused and within page limits. These supplementary results serve as supporting evidence rather than primary demonstrations of the framework.

---

> ### Author Response · Authors · 2025-11-22
>
> ---
> Q2.1:
> **Assumptions**:
> **A:** Our framework relies on a small set of general assumptions:
> - The environment admits a forward-dynamics function (e.g., Euler integration) that can be used to predict future object states under candidate actions.
>
>
> - An LLM can serve as a high-level symbolic planner operating on structured representations within an embodied AI system.
>
>
> - The perception module provides sufficiently accurate object states such that moderate noise can be tolerated through smoothing, relational filtering, and closed-loop control.
>
>
>
> ---
> Q2.2
> **“Second-order dynamics.”**
> **A:** Indeed, predicting second-order effects (e.g., interactions caused by living agents or man-made objects applying non-passive forces) is challenging for both simulators and world models.
> - For short-horizon tasks (seconds), rigid-body forward models remain accurate enough.
> - For longer horizons or environments with active agents, our framework allows replacing the simulator with learned world models or hybrid predictors.
> Thus, the choice of physical predictor is modular.
> ---
> Q2.3
> **”Why Graph models.”**
> **A&D:**
> This question touches on one of the central conceptual motivations behind our work, and it naturally leads to a broader discussion that extends beyond the scope of a rebuttal.
> At present, it remains unclear whether existing scaling laws for large models can directly extend to real-world embodied settings. From an information-theoretic perspective, intelligence improves when a model becomes better at compressing the patterns it observes. In physical environments, however, the most efficient and most general compression of the world is physics itself: physical laws are inherently universal and perfectly generalizable. This viewpoint motivates the physics/simulator component of PGLPA.
> Regarding our architectural choice, we prefer a Perception–Graph–LLM pipeline over a fully end-to-end model for several reasons:
> 1. Engineering rationale:
> - It converts relational information into a structured form that can be cleanly expressed to the LLM.
>
>
> - It improves interpretability, enabling engineers to debug the system and explicitly control the LLM’s context window.
>
>
> - It reduces the training burden compared with a monolithic end-to-end model.
>
>
> It allows hierarchical control: the LLM plans at a slower time scale, while low-level controllers operate at high FPS.
>
>
> 2. Scientific rationale:
>
> A graph-based representation is better suited than transformer- or diffusion-style latent models for grouping objects and identifying interaction relations.
>
>
> We hypothesize that relational structure is more important than isolated object states for embodied reasoning.
>
>
> This connects back to the earlier philosophical question: whether scaling alone can solve embodied intelligence, or whether explicit structure—such as a physical compressor—remains necessary.
>
>
> Our design choice reflects this position: meaningful generalization in embodied settings requires both the statistical power of large models and the explicit structure provided by physics-grounded relational representations.
>
>
>
> ---
> Q2.4
> **“Baselines for Tetris and other experiments.”**
> **A:** As noted earlier, we do **not** position these experiments as benchmark competitions.
> Our aim is to demonstrate:
>
> - In Physics QA: “simulated rollouts enhance LLM physical reasoning.”
> - In Tetris: “predicted future states improve action evaluation.”
>
> Domain-specific solvers (MCTS, heuristics, etc.) are not comparable baselines for a *generalist physics-grounded planning* system.
> Appropriate baselines would be VLA- or VLM-style general planners.
>
> ---
>
> **“Pipeline format / consistency.”**
> 	The pipeline format is demonstrated at General Comment(part 1).
>
> In the main pages, physical QA and Tetris are **independent demonstrations** of two claims:
>
> 1. The simulator can improve LLM reasoning.
> 2. Predicted future states help in choosing feasible or advantageous actions.
>
> In Tetris, the “simulator” is a future-state machine computing height/holes/cleared lines for each action; it does not recommend an optimal solution but provides state rollouts for the LLM to evaluate.

---

### Official Review · Reviewer_ZBxi · 2025-11-04

**Soundness:** 2
**Presentation:** 2
**Contribution:** 2
**Rating:** 2
**Confidence:** 2

**Summary:**

The paper proposes APEX, a plug-and-play loop that (i) constructs a relational scene graph from two snapshots, (ii) applies a “difference-graph” attention module (a Graphormer variant) to identify salient interactions, (iii) enumerates candidate actions, (iv) performs forward simulation rollouts with a physics engine for each action, and (v) asks an LLM to select an action using textual rollout summaries. Experiments cover synthetic physics QA, a custom short-horizon Tetris setup, a dynamic obstacle-avoidance toy task, PHYRE, and a small “real-world” vignette.

**Strengths:**

- Pragmatic integration of a physics engine into an LLM loop; the overall system is easy to understand and replicate in spirit.

- Clear problem motivation: teaching LLM agents to rely on external physics tools rather than internalizing fragile physical heuristics is a sensible direction.

- Breadth of tasks (synthetic QA, games/toy control, PHYRE) demonstrates that the loop can be wired up across settings.

**Weaknesses:**

- limited methodological novelty
- single-step lookahead: the algorithmic core is a 1-step brute-force evaluation of enumerated actions; claims and tables about multi-step/higher-horizon complexity are not matched by controlled, implemented evidence.
- Tetris lacks standard hand-coded/Tetris-AI baselines (e.g., height/holes/bumpiness heuristics, MCTS with lookahead).
- Obstacle avoidance omits classic MPC/DWA/A*/RRT with the same simulator and budget.
- PHYRE relies on 10k uniformly random actions, effectively reducing to random search rather than demonstrating LLM-guided planning

**Questions:**

Can you precisely define (\Delta G) for heterogeneous edge types and features. How are ((G_t, G_{t+\Delta t})) fused in attention (encodings, positional terms, pairwise features)?

---

> ### Author Response · Authors · 2025-11-22
>
> W1.1: **“Limited methodological novelty.”**
> **A:** We clarify here how our approach differs from prior simulation-in-the-loop methods such as Mind’s Eye and PhysVLM.
>  Those works typically focus on one-shot static reasoning (e.g., predicting which ball lands first) and do not embed physics-forward prediction into a closed action-planning loop in SE(3).
>  In contrast, our work targets general task planning in dynamic environments, where the agent must reason about evolving states, possible interventions, and the feasibility of multi-step action sequences.
>  Moreover, we propose an explicit unified Perception → Graph → Physics → Language → Action framework, where:
> Perception provides structured object states;
>
>
> - A graph module filters interaction saliency;
>
>
> - Physics rollouts serve as a quantitative oracle;
>
>
> - An LLM performs high-level reasoning and plan selection;
>
>
> - Actions are grounded back into SE(3).
>
>
> To our knowledge, this explicit modular composition, with physics rollouts as a first-class component inside the LLM planning loop, has not been formalized in prior work.
>
>
> ---
>
> W1.2:
>  **“Single-step lookahead.”**
> **A:** The concern about multi-step reasoning, action space, and search policy relates directly to our design choice in this first-stage study.
>  Our current implementation uses a simplified search strategy (essentially 1-step evaluation over an enumerated action set) because the goal of this paper is to demonstrate the feasibility and usefulness of physics-grounded forward simulation inside an LLM planning loop, not to propose a new SOTA action-planning algorithm.
> Crucially, nothing in the framework restricts it to single-step lookahead: the action module is designed to be replaceable by more expressive controllers (symbolic policies, MPC-based planners, hierarchical skills, or learned low-level controllers), and the physics oracle can be called iteratively for multi-step rollouts.
>  In follow-up work, we will show experiments with richer parameterized action primitives and longer-horizon search, using the same PGLPA structure.
>
>
> ---
>
> W1.3:
> **“Lack of classical baselines for Tetris and Obstacle Avoidance.”**
> **A:** We intentionally did **not** frame these experiments as benchmark competitions.
> The purpose is not to compare against task-specific solvers (e.g., MCTS for Tetris, MPC/A*/RRT for navigation) but to show that our *general* physics-grounded planning loop can be instantiated across heterogeneous domains.
>
> A fair comparison for our method is *generalist planning systems* (e.g., VLA or VLM-based planners), not hand-engineered domain-specific methods.
>
> ---
>
> W1.4:
> **“PHYRE reduces to random search rather than LLM-guided planning.”**
> **A:** We agree that PHYRE’s action space is extremely large and that many PHYRE tasks are weakly NP-hard, so a searching policy is naturally suitable for these tasks.
> Our intention is not to outperform specialized search heuristics, but to show that **physics rollouts can guide LLM physical reasoning**.
>
> ---
>
> Q1.1 **“Define ΔG and clarify fusion of (G_t, G_{t+Δt}).”**
> **A:** We will provide detailed definitions, heterogeneous edge handling, and attention fusion mechanisms in the updated Appendix.

---

### Author Response · Authors · 2025-11-22
**General Comment (part 1)**

We thank the reviewers for their careful reading and constructive feedback, which helped us improve the clarity and completeness of the paper.
### 1. **Contributions and Pipeline**

In this paper, we present two contributions.

**(1) LLM-based physical planning can be grounded through physical forward simulation**

Prior works such as Mind’s Eye[1] and related simulation-in-loop methods also use simulators, but they differ in two fundamental ways:
They focus on one-shot static reasoning (e.g., “which ball lands first”), whereas our focus is general planning in dynamic environments, where the agent must reason over evolving states and possible interventions.


They do not incorporate physics-forward prediction within an action planning loop. PhysVLM[2] is related, but it still focuses on static object questioning and does not perform Euler-style forward prediction of future states under candidate actions.
The three experiments (PhysQA, Tetris, Dynamic Obstacle Avoidance) are presented progressively to illustrate how APEX works across increasingly dynamic scenarios. Our purpose is not to “score” these benchmarks, but to show that such a pipeline can be instantiated and that the components required by PGLPA are operational.

**(2) PGLPA is a general embodied AI framework**

Our second contribution is PGLPA, a general embodied-AI formulation that unifies perception, structured scene representations, language-based reasoning, and physics-grounded rollout evaluation. The modular components (e.g., graph encoder and physics model) are designed to preserve explicit SE(3) structure and reduce the dependence on large training datasets.
In the remainder of the rebuttal, we focus on clarifying PGLPA and adding additional experiments that highlight its reasoning structure and capability.

### **Problem formulation**
Let $W$ denote a partially observed world, governed by physical laws $P$ and task constraints $C$.
A general agent must complete a goal $g$ under the constraints $(C, P)$.

---

### **Planning phase**

Given the current state $\{W, P\}$ and target $\{g, C\}$, the agent seeks a policy $p$, executed through actuators $A$, that produces a planned trajectory:

$s^{\text{plan}}_{t+k} \in (C, P)$ for $0 < k \le h$,

such that the terminal condition holds:

$s^{\text{plan}}_{t+h} = s\_{\text{goal}}$.

Here $h$ is the temporal horizon required by $C$, and $(t, t+h)$ denotes the planning window.

---

### **Execution phase**

The policy $p$ is then physically actuated by $A$, generating a real-world trajectory:

$s_{t+k} \in (C, P)$ for $0 < k \le h$,

and satisfying:

$s_{t+h} = s_{\text{goal}}$.

Both phases operate under the same temporal and physical constraints.

---

### **Task assumption**

In this work, we consider general planning in a known environment.
Models are pretrained and evaluated zero-shot on unseen scenes, but the agent already has an internal reconstruction of the environment from partial observations (for example, it already knows where the target object is; it does not need to search across rooms or infer whether an object exists).
In this setting, RL based methods are not suitable for general tasks.

---

## **Data pipeline of PGLPA**

### **Perception module**

**Input:** raw environment observations (e.g., visual or simulated).
**Output:** structured object states

$obj\_{states} = (shape_i, pos\_i, vel\_i, m_i, \mu\_i) $,

where $m_i$ (mass) and $\mu_i$ (friction coefficient w.r.t. the actuator) are estimated or assumed.

---

### **Graph module**

**Input:** $ obj_{states} $.
**Output:** an environment summary graph encoding:
- salient interactions (e.g., collisions, harmful contacts),
- group-level abstractions (“randomly posed cans”),
- group completion or reasoning (“a hand” $\rightarrow$ “a human”).

---

### **LLM**

**Input:** environment summary (and optionally the physics summary).
**Output:** a high-level plan expressed in natural language or structured text.

---

### **Physics module**

**Input:** $obj_{states}$ and the proposed plan.
**Output:** a physics summary: simulated rollouts, predicted outcomes, temporal consistency, final states.

---

### **Action execution**

**Input:** the selected plan.
**Output:** actuator-level commands executed in the environment.

---

### **Remarks**

This pipeline differs from the simplified version shown in Appendix 6.1.
When the action space is small and discrete, exhaustive search is feasible.
However, the purpose of this paper is **not** to build the full embodied system,
but to validate whether **physically grounded outcomes** can effectively guide a large language model during planning.


[1] Liu, Ruibo, et al. "Mind's eye: Grounded language model reasoning through simulation." arXiv preprint arXiv:2210.05359 (2022).

[2] Zhou, Weijie, et al. "Physvlm: Enabling visual language models to understand robotic physical reachability." Proceedings of the Computer Vision and Pattern Recognition Conference. 2025.

---

### Author Response · Authors · 2025-11-22
**General Comment (part 2)**

## 2. Action Space and Search Policy

Several concerns were raised about the simplicity of the action search component.
This is tightly connected to the core aim of our work:
**“demonstrating that physics-grounded forward simulation can be integrated into the action-planning loop of large models in SE(3) space.”**

In the main paper, we intentionally used a simplified, emulator-style action set and performed rollouts in a simulator.
This choice is **not** because we view exhaustive search as the optimal policy, but because our goal in this stage is purely **to verify the feasibility and utility of physics-in-the-loop planning**.

Importantly, the action module in our framework is **modular by design**.
It can be replaced by any compatible controller and grounded by physical models, including:

- Symbolic policies
- Classical motion planners
- Learned low-level controllers
- Hierarchical or options-based policies

For a follow-up work, we will include additional experiments with a more expressive and structured action space (e.g., parameterized skills, low-level controllers), demonstrating that the physics-LLM loop remains applicable under richer action parameterizations.

---

## 3. Rigid-Body Environments and Beyond

Some questions relate to soft-body, deformable, or liquid environments.
We agree this is an open research direction for *all* modeling approaches, including world models and simulators.

Our core claim is:

- **Given complete state information for a rigid-body environment, a physics simulator is strictly more accurate, efficient, and reliable than a learned world model.**
  This is because learned models approximate physics, whereas rigid-body simulation *implements* physics.

We do **not** reject world models; in fact, our framework allows the physics module to be replaced by any compatible predictor (learned or numerical).
The choice of model is orthogonal to the framework.

For soft or liquid environments, there are only two possible outcomes in the long term:

1. **A world model or improved simulator achieves quantitative accuracy**
   → We can plug it directly into our physics module.

2. **Neither learned nor numerical models can reliably predict such environments**
   → In that case, the agent must rely on **vision + force feedback** at execution time (i.e., closed-loop control), and PGLPA gracefully falls back to action-level adaptation.

In either scenario, the framework remains valid; the open question lies only in the choice of the “physics oracle.”

---

## 4. Additional Experiments (Planned Additions)

We will include 3 new experiments to further clarify the framework’s capability:

- **Third-party collision intervention**
Demonstrates zero-shot predictive physical reasoning in a scenario that is rarely covered by typical training data.

-  **Overheavy box picking up**
Shows how physics-based forward simulation prevents infeasible or unsafe actions (e.g., the planner recognizing insufficient gripping force).

- **LIBERO-PRO benchmark[3]**
Compare PGLPA with current VLA model on disturbed environment.

These additional experiments directly address concerns about generality, physical grounding, and real-world feasibility.

[3] Zhou, Xueyang, et al. "LIBERO-PRO: Towards Robust and Fair Evaluation of Vision-Language-Action Models Beyond Memorization." arXiv preprint arXiv:2510.03827 (2025).

---

### Note · Authors · 2025-12-03

I have read and agree with the venue's withdrawal policy on behalf of myself and my co-authors.